# CDK1 controls CHMP7-dependent nuclear envelope reformation

Alberto T Gatta[1,2†], Yolanda Olmos[1,2†‡], Caroline L Stoten[1,2†], Qu Chen[3], Peter B Rosenthal[4], Jeremy G Carlton[1,2]*

[1]School of Cancer and Pharmaceutical Sciences, King's College, London, United Kingdom; [2]Organelle Dynamics Laboratory, The Francis Crick Institute, London, United Kingdom; [3]Structural Biology Science Technology Platform, The Francis Crick Institute, London, United Kingdom; [4]Structural Biology of Cells and Viruses Laboratory, The Francis Crick Institute, London, United Kingdom

**Abstract** Through membrane sealing and disassembly of spindle microtubules, the Endosomal Sorting Complex Required for Transport-III (ESCRT-III) machinery has emerged as a key player in the regeneration of a sealed nuclear envelope (NE) during mitotic exit, and in the repair of this organelle during interphase rupture. ESCRT-III assembly at the NE occurs transiently during mitotic (M) exit and is initiated when CHMP7, an ER-localised ESCRT-II/ESCRT-III hybrid protein, interacts with the Inner Nuclear Membrane (INM) protein LEM2. Whilst classical nucleocytoplasmic transport mechanisms have been proposed to separate LEM2 and CHMP7 during interphase, it is unclear how CHMP7 assembly is suppressed in mitosis when NE and ER identities are mixed. Here, we use live cell imaging and protein biochemistry to examine the biology of these proteins during M-exit. Firstly, we show that CHMP7 plays an important role in the dissolution of LEM2 clusters that form at the NE during M-exit. Secondly, we show that CDK1 phosphorylates CHMP7 upon M-entry at Ser3 and Ser441 and that this phosphorylation reduces CHMP7's interaction with LEM2, limiting its assembly during M-phase. We show that spatiotemporal differences in the dephosphorylation of CHMP7 license its assembly at the NE during telophase, but restrict its assembly on the ER at this time. Without CDK1 phosphorylation, CHMP7 undergoes inappropriate assembly in the peripheral ER during M-exit, capturing LEM2 and downstream ESCRT-III components. Lastly, we establish that a microtubule network is dispensable for ESCRT-III assembly at the reforming nuclear envelope. These data identify a key cell-cycle control programme allowing ESCRT-III-dependent nuclear regeneration.

***For correspondence:**
jeremy.carlton@kcl.ac.uk

†These authors contributed equally to this work

**Present address:** ‡Department of Cell Biology, Universidad Complutense de Madrid, Madrid, Spain

**Competing interests:** The authors declare that no competing interests exist.

## Introduction

During division, cells undergo a programmed reorganisation of their interphase architecture to allow cyto- and karyokinesis. The division phase (M-phase) is set through the actions of key protein kinases such as Cyclin-Dependent Kinase-1 (CDK1) that modify the behaviour of proteins to orchestrate cellular reorganisation (*Champion et al., 2017*). To grant the spindle access to duplicated chromatids, the nuclear envelope (NE) is disassembled and is incorporated into the Endoplasmic Reticulum (ER) creating a hybrid organelle containing peripheral and transmembrane proteins from both organelles. As cells exit M-phase, they must regenerate both their nuclei and their NE. Here, membranes from the ER/NE hybrid organelle envelope nascent daughter nuclei, INM proteins recover their NE localisation, spindle microtubules traversing the NE are disassembled and remaining holes in the NE are sealed (*Carlton et al., 2020*). In addition to its roles in membrane trafficking and cytokinesis (*Vietri et al., 2020a*), ESCRT-III has emerged as a key orchestrator of nuclear envelope sealing during open mitosis (*Olmos et al., 2015*; *Vietri et al., 2015*; *Olmos et al., 2016*; *Gu et al., 2017*), the repair of ruptured micro- and parental nuclear envelopes in interphase (*Willan et al., 2019*;

*Raab et al., 2016*; *Denais et al., 2016*; *Robijns et al., 2016*; *Vietri et al., 2020b*) and the surveillance of damaged NPC complexes in *S. cerevisiae* (*Webster et al., 2016*; *Webster et al., 2014*). ESCRT-III is a membrane remodelling filamentous polymer that works *in-concert* with a AAA-ATPase called VPS4 that provides energy for filament remodelling. VPS4 is recruited to ESCRT-III proteins through engagement of sequences called MIMs (MIT-domain interaction motifs) through its MIT (Microtubule Interaction and Trafficking) domain found within ESCRT-III proteins. In the context of NE reformation, during M-exit, the inner nuclear membrane protein LEM2 assembles into a phase-separated gel-like polymer that defines sites of ESCRT-dependent nuclear envelope sealing through its ability to recruit and activate polymerisation of the ER-localised ESCRT-III protein, CHMP7 (*Gu et al., 2017*; *von Appen et al., 2020*). Furthermore, CHMP7 and LEM2 may also regulate nuclear envelope sealing by feeding new ER membrane, as recently shown in *C. elegans* (*Penfield et al., 2020*). In worms and fission yeast, LEM2 also has important roles in stabilising peripheral heterochromatin and in organising chromatin architecture in the interphase nucleus (*Ikegami et al., 2010*; *Barrales et al., 2016*; *Pieper et al., 2020*). Given the key role for LEM2 in stimulating CHMP7 polymerisation during M-exit, we wondered how the biology of CHMP7 and LEM2 was controlled during M-phase.

## Results

### CHMP7 is required for dissolution of LEM2 clusters that form during nuclear envelope regeneration

CHMP7 and LEM2 exhibit complex domain architectures (*Figure 1A*) and in interphase are localised to the ER and the INM, respectively. We took RNA-interference (*Figure 1B*) and stable cell line (*Figure 1C*) approaches to explore the CHMP7/LEM2 axis during M-exit. In early mitosis, GFP-CHMP7 and LEM2-mCh co-localised in a hybrid ER/NE, were then transiently co-enriched at sites of ESCRT-III assembly at the reforming NE, and subsequently adopted distinct ER and NE identities as cells exited mitosis (*Figure 1C*, *Figure 1—video 1*). As expected (*Gu et al., 2017*), LEM2-depletion prevented the transient assembly of CHMP7 at the reforming NE (*Figure 1—figure supplement 1*, *Figure 1—video 2*). We generated cells stably expressing LEM2-mCh or LEM2$^{\delta LEM}$-mCh (*Figure 1—figure supplement 2A*) and found that the LEM-domain was necessary for efficient LEM2 and CHMP7 enrichment at the reforming nuclear envelope, and subsequent retention of LEM2 at the NE (*Figure 1—figure supplement 2B* and *Figure 1—video 3*). In cells co-expressing GFP-CHMP7 and LEM2$^{\delta LEM}$-mCh, both proteins instead co-assembled in the peripheral ER after nuclear envelope reformation (*Figure 1—figure supplement 2C*). Deletion of the CHMP7-interaction domain from LEM2 (LEM2$^{\delta 415-485}$-mCh, *von Appen et al., 2020*) also resulted in the generation of clusters of LEM2 and nuclear envelope morphology defects in the following interphase (*Figure 1—figure supplement 2D*). These data suggest that LEM2 must effectively gain chromatin tethers during mitotic exit to localise CHMP7 to the reforming NE, and that persistent LEM2 in the peripheral ER can seed inappropriate CHMP7 clustering during mitotic exit. In *S. japonicus*, Cmp7 disaggregates Lem2 clusters during interphase and mitotic exit (*Pieper et al., 2020*; *Lee et al., 2020*). Here, we found that CHMP7 depletion led to persistent enrichment of LEM2 at the reforming NE (*Figure 1D*, *Figure 1E* and *Figure 1—video 4*), suggesting that CHMP7 acts additionally to disassemble LEM2 during M-exit. Importantly, depletion of IST1, an ESCRT-III subunit with well-characterised roles in nuclear envelope sealing (*Vietri et al., 2015*), did not lead to persistent LEM2 clusters (*Figure 1—figure supplement 3*), which is consistent with recent findings from yeast and worms suggesting functional diversity across ESCRT-III components during NE reassembly (*Penfield et al., 2020*; *Pieper et al., 2020*). Persistent assembly of LEM2 in CHMP7-depleted cells led to the formation of NE morphology defects during cytokinesis that persisted into the subsequent interphase (*Figure 1F* and *Figure 1G*). These LEM2 clusters did not incorporate other INM proteins such as LAP1 or Emerin (*Figure 1—figure supplement 4A*). VPS4 was neither recruited to the reforming NE in the absence of CHMP7, nor recruited to the LEM2 clusters induced by CHMP7-depletion (*Figure 1—figure supplement 4B–C*), suggesting that the clusters may form as a consequence of impaired remodelling of ESCRT-III. In cells bearing persistent LEM2 clusters, we observed a breakdown in nucleocytoplasmic compartmentalisation (*Figure 1H*) and the localisation of DNA-damage markers to the site of these clusters, suggesting that their formation is detrimental to cellular health (*Figure 1—figure*

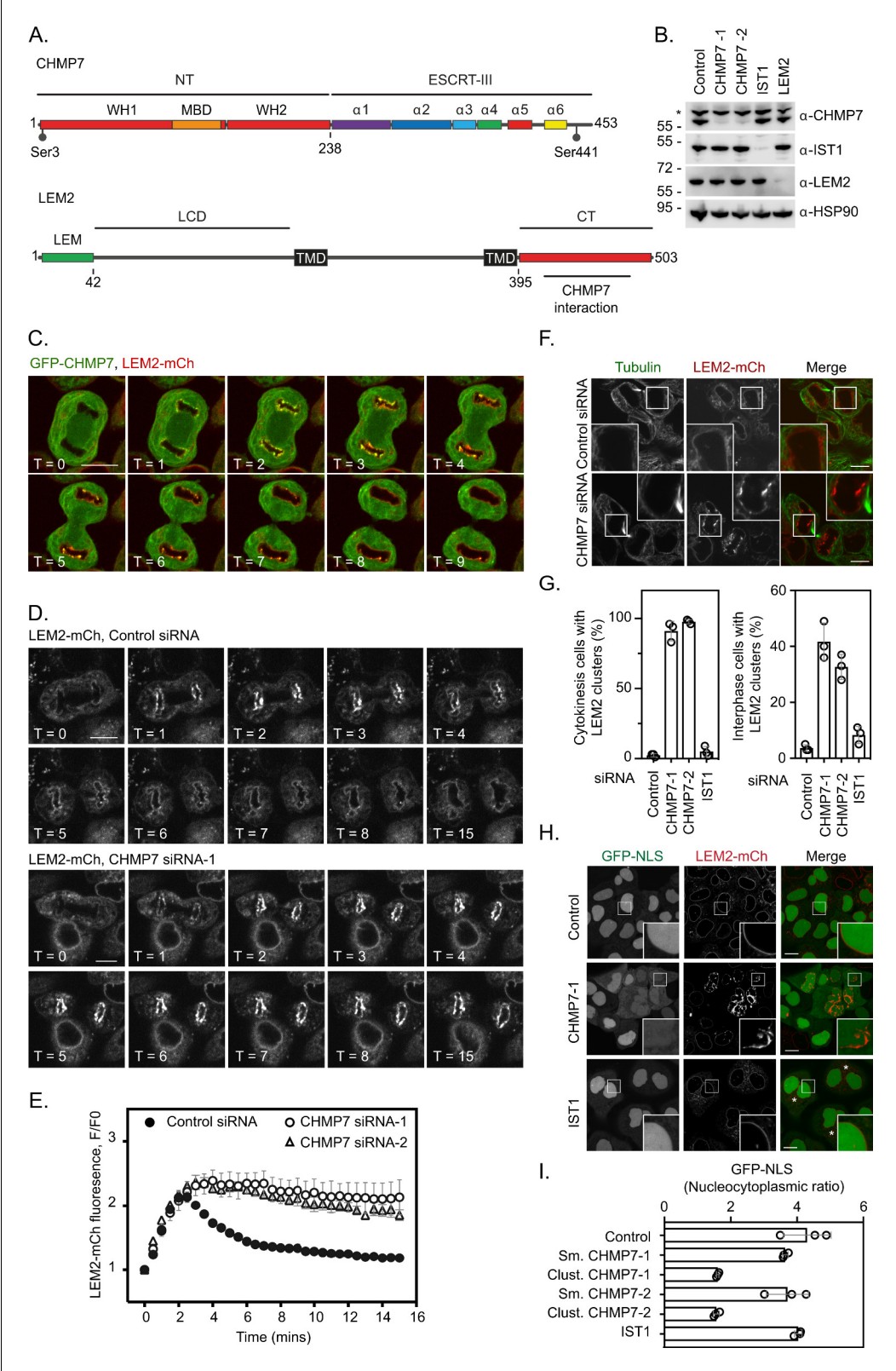

**Figure 1.** ESCRT-III controls LEM2 dynamics during nuclear envelope reformation. (**A**) Cartoon depicting CHMP7 and LEM2 secondary structural elements. CHMP7 contains an ESCRT-II like N-terminus (NT) comprising tandem Winged Helix (WH) domains and a Membrane Binding Domain (MBD) required for ER localisation. Mitotically phosphorylated Serine residues identified in this manuscript are highlighted. LEM2 comprises an N-terminal LEM domain, a low-complexity domain (LCD) that has been shown to promote phase separation, two transmembrane (TM) domains that result in its

*Figure 1 continued on next page*

*Figure 1 continued*

insertion into the INM, and a C-terminal (CT) WH domain containing a region required for interaction with CHMP7 (415–485). (**B**) Resolved lysates from HeLa cells that had been transfected with the indicated siRNA were examined by western blotting with antibodies raised against CHMP7, IST1, LEM2, or HSP90, asterisk marks non-specific band. (**C**) HeLa cells stably expressing both GFP-CHMP7 and LEM2-mCh were imaged live during mitotic exit. Images representative of 4/4 (GFP-CHMP7, LEM2-mCh) Videos analysed. (**D, E**) HeLa cells stably expressing LEM2-mCh were treated with Control or CHMP7-targetting siRNAs and imaged live during mitotic exit. LEM2-mCh fluorescence levels in a region of interest (ROI) around the chromatin discs were quantified in E from 25 (Control) or 18 (CHMP7 siRNA-1) or 14 (CHMP7 siRNA-2) individually imaged cells acquired over three independent experiments, mean ± S.E.M. presented. Significance was calculated using a one-way ANOVA, with Dunnett's post-hoc test, $p < 0.0001$ for both CHMP7 oligos compared to control siRNA. (**F, G**) HeLa cells stably expressing LEM2-mCh were treated with Control or CHMP7-targetting siRNA, fixed and processed for immunofluorescence. The number of cells displaying aberrant LEM2 clusters during cytokinesis or in interphase were quantified from 300 cells per condition, examined over three independent experiments and presented as mean ± S.E.M. (**G**). Significance was calculated using a 1-way ANOVA, with Dunnett's post-hoc test; CHMP7 siRNA-1 $p < 0.0001$ (interphase), $p = < 0.0001$ (cytokinesis); CHMP7 siRNA-2 $p = 0.0001$ (interphase), $p = < 0.0001$ (cytokinesis); IST1 siRNA $p = 0.456$ (interphase, n.s.), $p = 0.765$ (cytokinesis, n.s.). In all panels, time in minutes and a scale bar of 10 μm are reported. (**H, I**) HeLa cells stably expressing GFP-NLS and LEM2-mCh were transfected with the indicated siRNA and imaged live. The nucleocytoplasmic ratio was quantified from cells treated with the indicated siRNA from three independent experiments and presented as mean ± S.E.M. (control, 295 cells; CHMP7 siRNA-1, 315 cells; CHMP7 siRNA-2, 195 cells; IST1 siRNA 215 cells). In CHMP7-depleted cells, the population was split into those cells displaying smooth LEM2 at the NE (Sm.) or those in which the LEM2 was clustered (Clust.). Significant breakdown in nucleocytoplasmic compartmentalisation was observed in the population of CHMP7-depleted cells displaying clusters ($p = <0.0001$ for CHMP7 siRNA-1 or CHMP7 siRNA-2 compared to controls, one-way ANOVA, with Dunnett's post-hoc test).

The online version of this article includes the following video, source data, and figure supplement(s) for figure 1:

**Source data 1.** CHMP7-dependent dissolution of LEM2 clusters during M-exit.

**Figure supplement 1.** LEM2 is needed for CHMP7 recruitment to the reforming NE.

**Figure supplement 2.** Activities within LEM2 controlling its assembly at the nuclear envelope.

**Figure supplement 2—source data 1.** Nuclear envelope morphology defects in the presence of LEM2 lacking the CHMP7-binding region.

**Figure supplement 3.** IST1 is dispensable for LEM2 cluster dissolution during M-exit HeLa cells stably expressing LEM2-mCh were treated with Control or IST1-targetting siRNA and imaged live during mitotic exit, as per *Figure 1E*.

**Figure supplement 3—source data 1.** IST1 is dispensible for dissolution of LEM2 clusters during M-exit.

**Figure supplement 4.** Analysis of proteins incorporated into LEM2-clusters formed in the absence of CHMP7.

**Figure supplement 5.** Effect of LEM2 clusters on genome integrity and organisation.

**Figure supplement 5—source data 1.** Affect of CHMP7 depletion on the formation of heterochromatin-1 clusters.

**Figure 1—video 1.** Related to *Figure 1C*.

https://elifesciences.org/articles/59999#fig1video1

**Figure 1—video 2.** Related to *Figure 1—figure supplement 1B*.

https://elifesciences.org/articles/59999#fig1video2

**Figure 1—video 3.** Related to *Figure 1—figure supplement 2C*.

https://elifesciences.org/articles/59999#fig1video3

**Figure 1—video 4.** Related to *Figure 1D and E*.

https://elifesciences.org/articles/59999#fig1video4

---

*supplement 5A*). In *S. japonicus*, Lem2 clusters generated in the absence of ESCRT-III sequester heterochromatin (*Pieper et al., 2020*). We wondered if persistent LEM2 aggregation during mitotic exit may perturb chromatin structure in CHMP7-depleted mammalian cells. We followed endogenous Heterochromatin Protein 1 (HP1) after CHMP7 depletion and instead found diminished HP1 foci in these cells (*Figure 1—figure supplement 5B–C*). This defect was more pronounced in cells exhibiting LEM2-mCh clusters (*Figure 1—figure supplement 5D*) and whilst there were a few remaining HP1 clusters in CHMP7-depleted cells, these did not co-localise with LEM2 (*Figure 1—figure supplement 5E*). Thus, as well as roles in spindle disassembly (*Vietri et al., 2015*; *von Appen et al., 2020*) and membrane fusion (*Vietri et al., 2015*; *Olmos et al., 2016*; *Gu et al., 2017*), we suggest that CHMP7 also controls a separate process of LEM2 dissolution during mitotic exit to allow both generation of a normal nuclear envelope and proper organisation of heterochromatin during the following interphase.

## Multiple nuclear export sequences (NESs) in CHMP7's C-terminus ensure its nuclear exclusion

We reasoned that spatiotemporal regulation of the LEM2/CHMP7 interaction would be essential both during mitotic exit and in the subsequent interphase. CHMP7 contains predicted Nuclear

Export Sequences (NESs) in Helices 5 and 6 that are thought to limit its exposure to INM proteins, including LEM2, during interphase (*Vietri et al., 2020b*; *Thaller et al., 2019*). We confirmed the presence of active NESs in helices 5 and 6 of CHMP7 (*Figure 2A* and *Figure 2—figure supplement 1A* to 1C) and demonstrated that transient perturbation of exportin function resulted in nuclear accumulation of GFP-CHMP7 (*Figure 2—figure supplement 1D and E*). Transient transfection of NES-compromised CHMP7 constructs led to sequestration of LEM2-mCh in nuclear and cytosolic clusters during interphase, with an additive effect observed upon disrupting both NESs (*Figure 2B* and *Figure 2—figure supplement 1F*). We were unable to generate cells constitutively expressing GFP-CHMP7$^{NES1-\&NES2-}$, however, by transient transduction of cells with retroviruses driving weak GFP-CHMP7$^{NES1-\&NES2-}$ expression, we could demonstrate the inappropriate capture of LEM2 during interphase in GFP-CHMP7$^{NES1-\&NES2-}$-positive nuclear envelope and cytoplasmic clusters (*Figure 2—figure supplement 1G*). These data highlight the essential nature of the trans-nuclear envelope segregation of LEM2 and CHMP7, suggesting the existence of a surveillance system poised to monitor the integrity of this barrier (*Vietri et al., 2020b*; *Thaller et al., 2019*) and showing that if this segregation is compromised then LEM2 and CHMP7 can inappropriately cluster.

The second NES in Helix6 overlaps with a predicted type-1 MIM (*Schöneberg et al., 2017*, *Figure 2A*), suggesting that this region may regulate both nucleocytoplasmic compartmentalisation and the ability of CHMP7 to engage the AAA-ATPase, VPS4. While deletion of Helix6 mimicked abrogation of Helix6's NES by elevating CHMP7's nucleoplasmic localisation, it also induced the appearance of CHMP7 clusters at the NE and in the peripheral ER (*Figure 2C and D*). We wondered if this clustering was due to impaired CHMP7 disassembly due to compromised interaction with VPS4. However, we found that CHMP7's C-terminus was unable to bind the isolated VPS4 MIT domain, and careful analysis revealed that charge substitution at two critical acidic residues has inactivated the predicted type-1 MIM in CHMP7's Helix6 (*Figure 2A* and *Figure 2—figure supplement 2*).

## Spatially inappropriate assembly of CHMP7 in the peripheral ER during M-exit sequesters ESCRT-III subunits and LEM2

We next followed cells expressing GFP-CHMP7$^{\delta Helix6}$ through mitotic exit. Here, we were surprised to find that this protein failed to assemble at the NE, but instead assembled in a timely, but spatially inappropriate manner in the peripheral ER (*Figure 2E*, *Figure 2—video 1*), sequestering both downstream ESCRT-III subunits, such as IST1 (*Figure 2—figure supplement 3A and B*), and LEM2-mCh (*Figure 2F*). These assemblies persisted into the next interphase, retaining incorporation of ESCRT-III components and LEM2-mCh and preventing LEM2 from adopting its normal interphase INM localisation (*Figure 2G*). Given the absence of regulated nucleocytoplasmic transport during this phase of mitosis, and the absence of a functional MIM in CHMP7 (*Figure 2—figure supplement 2*), these data argue against canonical VPS4-binding or the CHMP7-NESs as contributing to this spatially inappropriate assembly. Removal of the C-terminal regulatory region of ESCRT-III subunits is thought to convert them into an 'open' conformation, facilitating their polymerisation (*Shim et al., 2007*). We found that GFP-CHMP7$^{\delta Helix6}$ could more efficiently precipitate partner ESCRT-III subunits (*Figure 2—figure supplement 3C*), suggesting that like the described *S. cerevisiae* Chm7$^{open}$ (*Webster et al., 2016*), its autoinhibition has been relieved.

## Dissolution of CHMP7 clusters occurs upon M-entry and correlates with multisite phosphorylation of CHMP7 by CDK1

When analysing our time-lapse data, we noticed that interphase assemblies of GFP-CHMP7$^{\delta Helix6}$ were efficiently disassembled upon entry into the next mitosis, but reformed again during mitotic exit (*Figure 3A*, *Figure 3B* and *Figure 3—video 1*). Indeed, GFP-CHMP7 and GFP-CHMP7$^{\delta Helix6}$ were largely indistinguishable during early anaphase (*Figure 2F*). We wondered if there existed a 'reset' mechanism on the assembly status of CHMP7 during mitotic entry, to prepare it for spatiotemporally controlled polymerisation during mitotic exit. Interrogation of the Phosphosite database (*Hornbeck et al., 2014*) reveals that CHMP7 and LEM2 possess a number of annotated sites of post-translational modification. By capturing GFP-CHMP7 from interphase or mitotic cells, we discovered that CHMP7 was phosphorylated upon mitotic onset (*Figure 3C*). We mapped sites of mitotic phosphorylation to Ser3 in CHMP7's N-terminus and to Ser441 in CHMP7's C-terminus

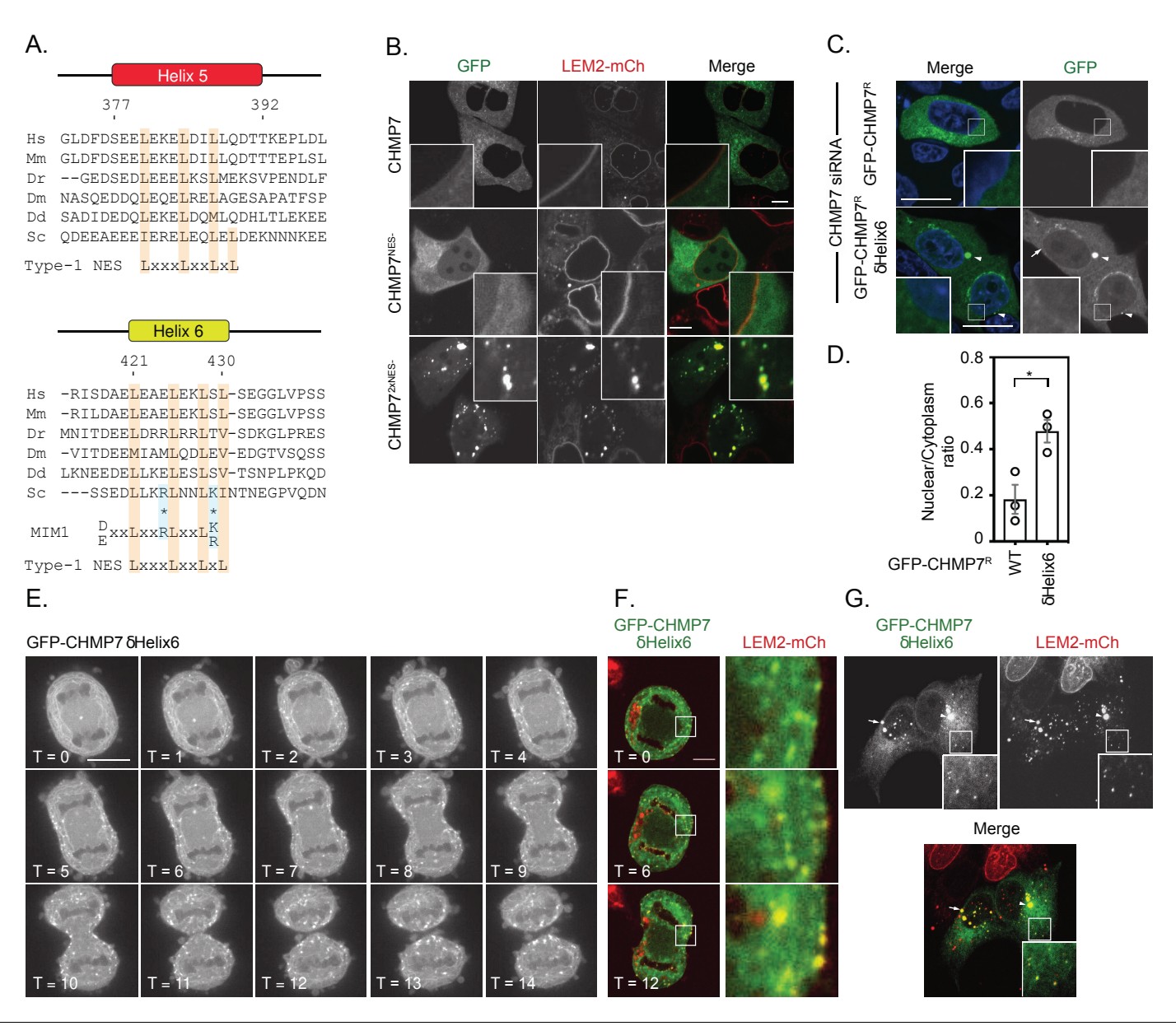

**Figure 2.** Deletion of a regulatory helix in CHMP7's C-terminus decouples its assembly from the nuclear envelope during mitotic exit and causes inappropriate sequestration of LEM2 in interphase. (A) Sequence alignment of CHMP7 Helix five and Helix six across phyla. Hs, *Homo sapiens*; Mm, *Mus musculus*; Dr, *Danio rerio*; Dm, *Drosophila melanogaster*; Dd, *Dictyostelium discoideum*; Sc, *Saccharomyces cerevisiae*. Type-1 Nuclear Export Sequence (NES) and MIM1 consensus sequence presented underneath; charge substituted residues in mammalian CHMP7 MIM1 indicated by asterisks. (B) HeLa cells stably expressing LEM2-mCh were transfected with plasmids encoding GFP-CHMP7, GFP-CHMP7 NES2-, GFP-CHMP7NES1- and NES2-, and imaged live. (C) HeLa cells transfected with CHMP7-targeting siRNA and transiently expressing GFP-CHMP7$^R$ or GFP-CHMP7$^R$ δHelix6 were fixed, stained with DAPI and imaged. GFP-CHMP7$^R$ δHelix6 was poorly exported from the nucleus, assembled aberrantly in the cytoplasm (arrowheads) and was not clearly enriched at the nuclear envelope (arrow). (D) Nucleocytoplasmic ratio of GFP-CHMP7 or GFP-CHMP7 δHelix6 from C. was quantified. Data presented as mean ± S.E.M from three independent experiments WT, n = 20, δHelix6, n = 117. Statistical significance was calculated with an unpaired two-tailed T-test, p = 0.023. (E) HeLa cells expressing GFP-CHMP7$^{δHelix6}$ were imaged live during mitotic exit. Extra-nuclear envelope clustering of GFP-CHMP7$^{δHelix6}$ was observed in 14/15 Videos acquired. (F, G) LEM2-mCh stable HeLa cells expressing GFP-CHMP7$^{δHelix6}$ were imaged live during mitotic exit (images representative of 4/4 Videos acquired) or fixed and the localisation of LEM2-mCh was determined. Removal of LEM2-mCh from the INM and its sequestration in clusters observed in 18/18 imaged cells.

The online version of this article includes the following video, source data, and figure supplement(s) for figure 2:

**Source data 1.** Nucleocytoplasmic distribution of GFP-CHMP7 or GFP-CHMP7 delta-Helix6.

**Figure supplement 1.** Nuclear export sequences (NESs) within CHMP7's C-terminus protect against unregulated exposure to LEM2.

*Figure 2 continued on next page*

*Figure 2 continued*

**Figure supplement 1—source data 1.** Examining NES sequences within CHMP7's C-terminus.
**Figure supplement 2.** CHMP7 Helix6 does not contain a functional MIM.
**Figure supplement 2—source data 1.** CHMP7's C-terminus does not bind to VPS4's MIT domain.
**Figure supplement 3.** Deletion of CHMP7's Helix6 results in its precocious assembly.
**Figure 2—video 1.** Related to *Figure 2E*.

https://elifesciences.org/articles/59999#fig2video1

($Figure\ 3D$, **Figure 3—figure supplement 1**). Both sites conform to consensus sequences ([K/H]-[pS]-[P] or [pS]-[P]-[X]-[R/K]) for the major mitotic kinase CDK1 and we could detect modification of these sites in immunoprecipitated samples from M-phase extracts using phospho-specific antibodies directed against these CDK1 substrate phosphorylation sites (**Figure 3C**).

## CDK1 phosphorylation of CHMP7 suppresses its assembly and inappropriate capture of LEM2 during M-exit

To understand how M-phase phosphorylation affected CHMP7's behaviour, we turned to a previously described sedimentation assay (**von Appen et al., 2020**). We found that recombinant CHMP7 could be directly phosphorylated by CDK1/CCNB1 in vitro and that CDK1-phosphorylation reduced the capacity of CHMP7 to sediment (**Figure 4A and B**). Importantly, incorporation of phosphomimetic residues at Ser3 and Ser441 (CHMP7 S3D/S441D and CHMP7 S3E/S441E) reduced this sedimentation (**Figure 4C and D**). These data suggest that the M-phase phosphorylations placed by CDK1 restrict the ability of CHMP7 to interact with itself and form sedimentable clusters.

We found that LEM2's C-terminus (LEM2$^{CT}$) could be co-sedimented with CHMP7 (**Figure 4—figure supplement 1A and B**), but its sedimentation with CHMP7 S3D/S441D and CHMP7 S3E/S441E was reduced (**Figure 4—figure supplement 1C and D**), suggesting that the CHMP7 assembly formed in these assays could interact with LEM2$^{CT}$. Given that CHMP7 alone may form non-specific aggregates which could pellet by sedimentation, we next used a sedimentation-independent interaction assay to test the effect of CHMP7 phosphorylation on its ability to bind LEM2$^{CT}$. Here, we discovered that whilst CHMP7 could interact well with LEM2$^{CT}$, either CDK1-phosphorylated CHMP7 or CHMP7 S3D/S441D were less able to interact with LEM2$^{CT}$ (**Figure 4E–H**). These data suggest that CDK1 phosphorylation of CHMP7 reduces both its ability to self-associate and its ability to bind LEM2. As the LEM2 interaction is thought to activate polymerisation of CHMP7, we next examined the nature of the LEM2/CHMP7 assemblies produced by negative-stain electron microscopy. We observed that LEM2$^{CT}$ induced polymerisation of CHMP7 in vitro (**Figure 4I**, **Figure 4—figure supplement 2**), producing curved filaments with periodicity matching the published dimensions and repeat units of the CHMP7 polymer observed previously (**von Appen et al., 2020**). Consistent with the reduced interaction between CHMP7 S3D/S441D and LEM2$^{CT}$ (**Figure 4G and H**), these filaments were not produced when LEM2$^{CT}$ was incubated with CHMP7 S3D/S441D (**Figure 4I**). In addition to curved filaments of CHMP7, we also observed curved lattices of CHMP7, suggesting that both linear and lateral polymeric interactions were induced (**Figure 4I**, **Figure 4—figure supplement 2**). These in vitro data support the hypothesis that CDK1-phosphorylation of CHMP7 acts to restrict its polymerisation by suppressing self-interaction and interaction with LEM2.

## CDK1-phosphorylation of CHMP7 persists beyond the timeframe nuclear envelope reassembly

We next used monopolar spindle assays (**Hu et al., 2008**) to examine the synchronised behaviour of CHMP7 during mitotic exit. Here, we synchronised cells at prometaphase using the Eg5 inhibitor STLC (**Skoufias et al., 2006**) and then employed RO-3306, a highly specific CDK1 inhibitor (**Vassilev et al., 2006**), to force synchronous mitotic exit. We used previously described CAL-51 cells in which the *CHMP7* locus had been homozygously edited to encode mNG-CHMP7 (**Olmos et al., 2016**). Following STLC arrest, we found that mNG-CHMP7 initiated assembly 16.32 ± 0.88 min after CDK1 inhibition and polymerised in a wave around the chromatin mass from the spindle-distal to the spindle-engaged face (**Figure 5A** and **Figure 5—video 1**). Assembly of endogenous CHMP7 persisted for 2.67 ± 0.22 min (**Figure 5C**), mimicking the assembly dynamics around the telophase

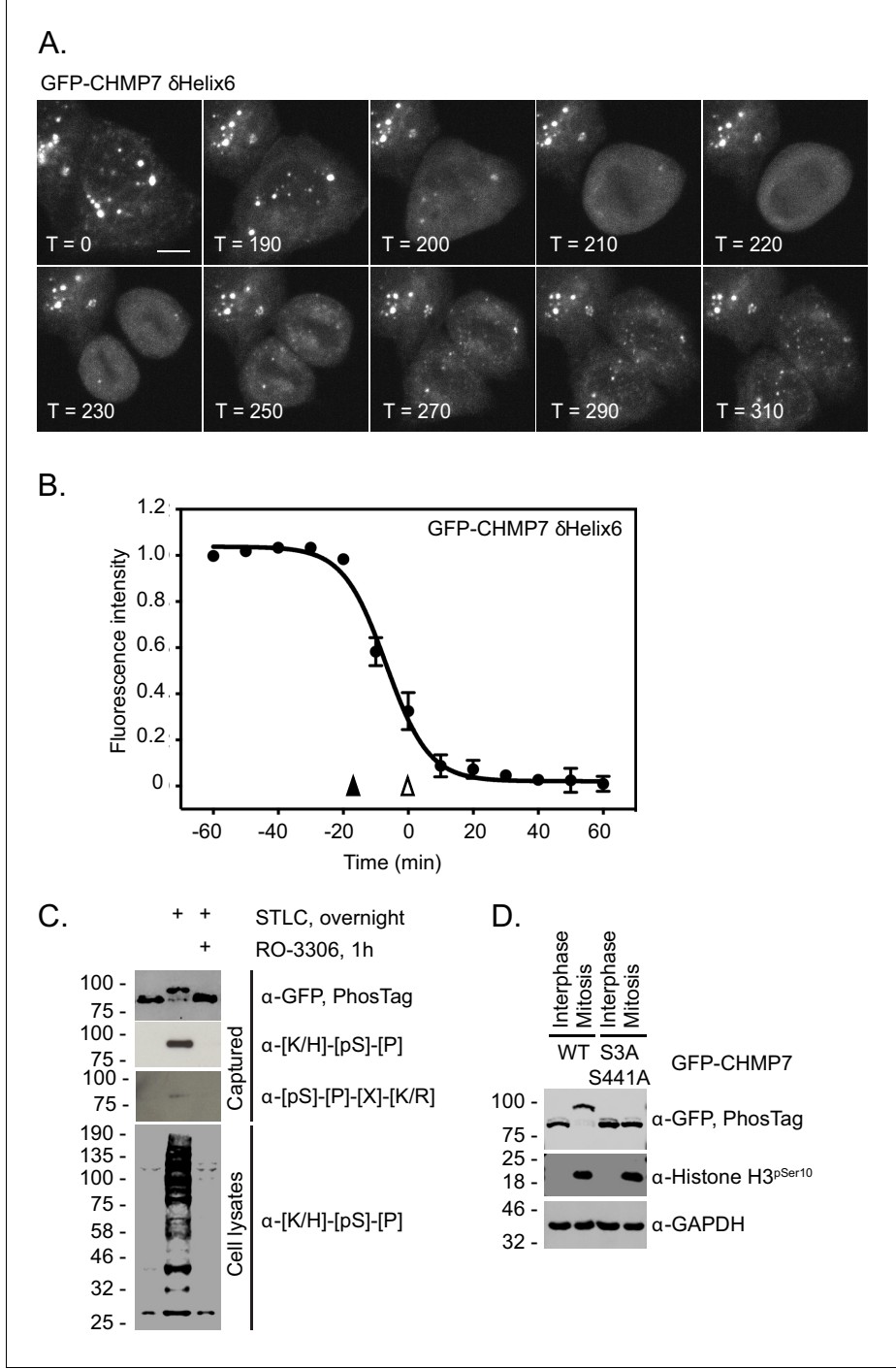

**Figure 3.** CHMP7 is phosphorylated by CDK1 upon mitotic entry. (**A, B**) HeLa cells expressing GFP-CHMP7 δHelix6 were imaged live (**A**) and the intensity of GFP-CHMP7 δHelix6 puncta was quantified during M-phase (**B**). Quantification of 59 cells from nine experiments. Fluorescence traces normalised to metaphase onset (open arrowhead); nuclear envelope breakdown indicated by closed arrowhead. (**C**) Lysates of HeLa cells stably expressing GFP-CHMP7 and subject to the indicated synchronisations were immunoprecipitated using GFP-trap resin and resolved using normal or PhosTag SDS-PAGE. Inputs and captured fractions were examined by western blotting with antisera raised against GFP, or CDK1 substrate consensus sequences ([K/H]-[pS]-[P] or [pS]-[P]-[X]-[R/K]). Western blots representative of 6 PhosTag immunoprecipitations. (**D**) Lysates of HeLa cells stably expressing GFP-CHMP7 or GFP-CHMP7 S3A, S441A and subject to the indicated synchronisations were immunoprecipitated using GFP-trap resin and resolved using normal or PhosTag SDS-PAGE. Inputs and captured fractions were examined by western blotting with antisera raised against GFP, Histone H3 pSer10 or GAPDH (N = 3).

*Figure 3 continued on next page*

*Figure 3 continued*

The online version of this article includes the following video, source data, and figure supplement(s) for figure 3:

**Source data 1.** GFP-CHMP7 delta-Helix6 cluster disassembly during M-phase.

**Figure supplement 1.** Mapping mitotic phosphorylation of CHMP7 to Ser3 and Ser441.

**Figure 3—video 1.** Related to *Figure 3A*.

https://elifesciences.org/articles/59999#fig3video1

nuclear envelope in asynchronous cells both in duration and in that CHMP7 in the peripheral ER was resistant to the assembly signal. In CHMP7-depleted HeLa cells stably expressing siRNA-resistant GFP-CHMP7 (GFP-CHMP7$^R$), CHMP7 assembled with similar kinetics, initiating 16.93 ± 1.21 min after CDK1 inhibition and persisting for 3.32 ± 0.13 min (*Figure 5—figure supplement 1A*, *Figure 5C* and *Figure 5—video 2*). Importantly, LEM2-mCh displayed similar kinetics, becoming enriched at the monopolar nuclear envelope 16.21 ± 2.23 min after CDK1 inhibition, and persisting for 4.11 ± 0.44 min. In contrast to CHMP7, LEM2-mCh localised more strongly at the spindle-engaged face of the NE (*Figure 5B–C* and *Figure 5—video 3*) before being depolymerised and maintaining NE localisation (*Figure 5B*). Enrichment at the spindle-engaged face is perhaps indicative of its phase-separated interaction with microtubules (*von Appen et al., 2020*). We next asked whether microtubules were necessary for CHMP7 assembly by using nocodazole instead of STLC to arrest cells at pro-metaphase. In this case, endogenous mNG-CHMP7 assembled and disassembled at the nuclear envelope with normal kinetics (*Figure 5C*, *Figure 5—figure supplement 1B* – 1D, *Figure 5—video 4*), suggesting that microtubules are dispensable for ESCRT-III assembly at this site. Interestingly, endogenous mNG-CHMP7 usually assembled synchronously around the perimeter of the chromatin mass (*Figure 5A*). However, in cells that were able to form a pseudo-furrow (*Hu et al., 2008*), CHMP7 assembled at the furrow-proximal face, before spreading to the rest of the chromatin, suggesting that there may be a furrow-directed spatial component to ESCRT-III dynamics at the reforming NE (*Figure 5—figure supplement 1B*, *Figure 5—video 5*). Consistent with experiments from cycling cells (*Figure 1*), in CHMP7-depleted cells stably expressing LEM2-mCh, LEM2-mCh was poorly disassembled after nuclear envelope enrichment (*Figure 5—figure supplement 1E* and *Figure 5—video 6*). These persistent LEM2-mCh assemblies did not colocalise with microtubules, suggesting that they do not form as a consequence of impaired spindle disassembly (*Figure 5—figure supplement 1F and G*). Biochemical analysis of whole cell lysates from these monopolar spindle assays revealed phosphorylation of total CHMP7 at both Ser3 and Ser441 persisted for 20 min following CDK1 inhibition, with dephosphorylation completing 30 min after CDK1 inhibition (*Figure 5D* and *Figure 5E*). Importantly, these data suggest that ESCRT-III-dependent nuclear envelope reformation occurs in a time period when the bulk of CHMP7 in the peripheral ER is phosphorylated, and that complete CHMP7 dephosphorylation only occurs after nuclear envelope reformation. We wondered if there were spatiotemporal differences in the dephosphorylation kinetics of CHMP7. Using biochemical fractionation, we found that only a minor pool of CHMP7 could be detected in a chromatin-associated fraction during M-exit (*Figure 5—figure supplement 2A*) and we discovered that CHMP7 in this pool was dephosphorylated in advance of the larger extra-chromatin ER and cytosolic pool (*Figure 5—figure supplement 2B* – 2C). We next raised an antibody that was able to detect pSer3 CHMP7 in mitotic cells and discovered that the pool of endogenous CHMP7 assembling at the reforming NE was not illuminated by this antibody (*Figure 5—figure supplement 2D* – 2I). These data suggest that phosphorylation of the bulk of CHMP7 in the peripheral ER persists during M-exit, whereas the pool of CHMP7 in contact with the reforming NE is dephosphorylated in advance.

## CHMP7 phosphorylation restricts precocious assembly during M-exit

We next used CHMP7-depletion in cells stably expressing RNAi-resistant versions of CHMP7 (CHMP7$^R$) to examine the role of this phosphorylation during M-exit. We found that CHMP7 bearing non-phosphorylatable residues at positions 3 and 441 (GFP-CHMP7$^R$ S3A/S441A) assembled at the NE with similar kinetics and intensity to GFP-CHMP7$^R$, but displayed limited precocious clustering on the peripheral ER during early M-phase (*Figure 5F–H*, *Figure 5—figure supplement 3A and B*). During telophase, these precocious clusters of GFP-CHMP7$^R$ S3A/S441A grew larger (*Figure 5H*,

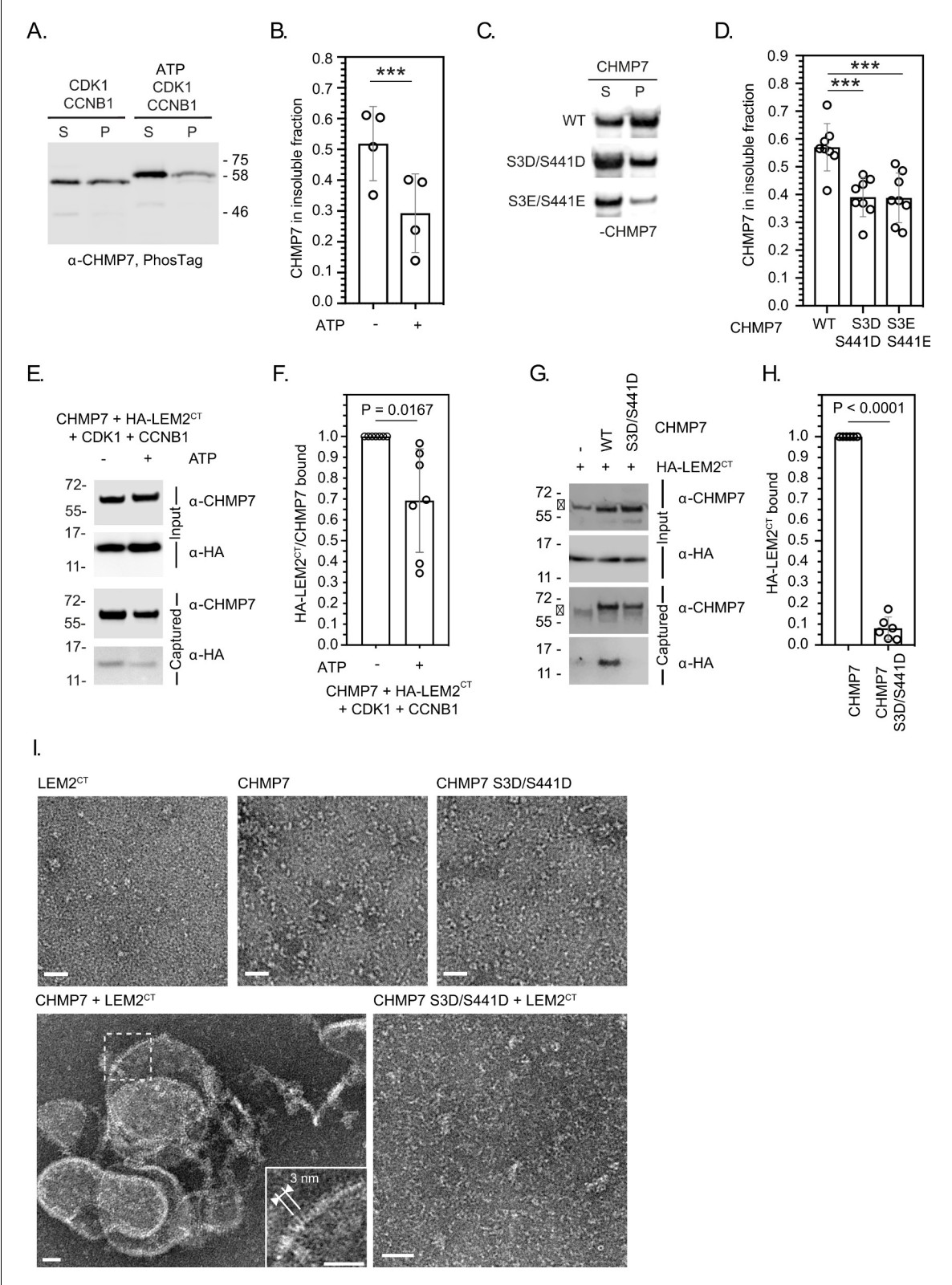

**Figure 4.** CDK phosphorylation suppresses CHMP7 sedimentation, interaction with LEM2 and LEM2-stimulated polymerisation. (**A**) Recombinant CHMP7 was incubated with recombinant CDK1 and CCNB1 in the presence or absence of ATP and sedimented by ultracentrifugation. CHMP7 was recovered from pellet and supernatant fractions, respectively, resolved by PhosTag SDS-PAGE and analysed by western blotting using antisera raised against CHMP7. (**B**) Data from A presented as mean ± S.D., N = 4, *** p = 0.0007 by paired two-tailed T-test. (**C**) Recombinant CHMP7, CHMP7 S3D/

*Figure 4 continued on next page*

*Figure 4 continued*

S441D, or CHMP7 S3E/S441E were sedimented by ultracentrifugation, recovered from pellet and supernatant fractions, and analysed by western blotting with antisera raised against CHMP7. (D) Data from C presented as mean ± S.D., N = 8, *** p = 0.0004. (E) Recombinant CHMP7 was incubated with recombinant CDK1 and CCNB1 in the presence or absence of ATP. Recombinant HA-LEM2$^{CT}$ was added and CHMP7 was captured on magnetic anti-CHMP7 Dynabeads. Captured and input fractions were resolved by SDS-PAGE and examined by western blotting with anti-CHMP7 or anti-HA antisera. (F) Data from G are presented as mean ± S.D., N = 7, p = 0.0156 by paired two-tailed T-test. (G) Recombinant HA-LEM2$^{CT}$ was incubated alone, with CHMP7 or CHMP7 S3D/S441D. CHMP7 was captured on magnetic anti-CHMP7 Dynabeads. Captured and input fractions were resolved by SDS-PAGE and examined by western blotting with anti-CHMP7 or anti-HA antisera. Please note non-specific bands detected by CHMP7 antibody (*). (H) Data from G are presented as mean ± S.D., N = 6, p < 0.0001 by paired two-tailed T-test. (I) Recombinant LEM2$^{CT}$, CHMP7, or CHMP7 S3D/S441D were examined by negative stain electron microscopy, either alone or in the combinations indicated. When incubated alone, no examples of polymerisation were observed. Images representative of N = three independent experiments. Scale bar is 25 nm.

The online version of this article includes the following source data and figure supplement(s) for figure 4:

**Source data 1.** Influence of CDK1-phosphorylation on the CHMP7/LEM2 interaction.
**Figure supplement 1.** CDK phosphorylation on CHMP7 Ser3 and Ser441 suppresses CHMP7 sedimentation and capture of LEM2$^{CT}$.
**Figure supplement 1—source data 1.** Examination of the CHMP7 polymer by negative stain EM.
**Figure supplement 2.** Electron microscopy of the CHMP7/LEM2$^{CT}$ polymer.

*Figure 5—video 7*) and similarly to the clusters induced by GFP-CHMP7$^{\delta Helix6}$ or LEM2$^{\delta LEM}$, incorporated ESCRT-III components and persisted into the next interphase (*Figure 5—figure supplement 3C*). These data suggest that M-phase phosphorylation of CHMP7 acts to suppresses its inappropriate assembly on the peripheral ER during both M-phase and M-exit. We next examined the behaviour of CHMP7 bearing phosphomimetic residues at position 3 and 441 (GFP-CHMP7$^R$ S3D/S441D) and were surprised to find that these mutations reduced, although did not eliminate, its assembly at the reforming NE (*Figure 5F–H*). Using the degree of cleavage furrow ingression at the point of maximal recruitment as a proxy for progression through M-phase, we found that the residual GFP-CHMP7$^R$ S3D/S441D was recruited later than WT CHMP7 (*Figure 5G*). These data suggest that incorporation of phosphomimetic residues at Ser3 and Ser441 limits its assembly at the reforming NE.

## Precocious assembly of CHMP7 compromises LEM2 during interphase

We next looked to rescue the LEM2 clustering and nucleocytoplasmic compartmentalisation phenotypes in cells stably expressing both LEM2-mCh and GFP-NLS using siRNA-resistant HA-tagged versions of CHMP7. Consistent with a limited assembly of GFP-CHMP7$^R$ S3D/S441D at the reforming NE, we found that like HA-CHMP7$^R$, HA-CHMP7$^R$ S3D/S441D, and HA-CHMP7$^R$ S3E/S441E could rescue both the clustering of LEM2 and the nuclear compartmentalisation breakdown induced by CHMP7 depletion (*Figure 6A and B*). Importantly, in CHMP7-depleted cells expressing HA-CHMP7$^R$ S3A/S441A, cytoplasmic and NE clusters of LEM2 persisted, and the nuclear compartmentalisation defect induced by CHMP7-depletion was only partially rescued (*Figure 6A–B* and *Figure 6—figure supplement 1A*). In CHMP7-depleted cells stably expressing both LEM2-mCh and GFP-CHMP7$^R$ S3A/S441A, LEM2-mCh was incorporated into precocious GFP-CHMP7$^R$ S3A/S441A-positive clusters that formed during M-exit (*Figure 6C*, *Figure 6—figure supplement 1B* and *Figure 6—video 1*). These clusters displayed fusogenic behaviour (*Figure 6—figure supplement 1C* and *Figure 6—video 2*) and long-term culture of cells bearing LEM2-mCh and GFP-CHMP7$^R$ S3A/S441A resulted in the loss of LEM2 from the INM with its retention in singular extra-nuclear clusters (*Figure 6—figure supplement 1D* and *Figure 6—video 2*). These data highlight CDK1-phosphorylation of CHMP7 as a mechanism to restrict polymerisation and precocious capture of LEM2 in the peripheral ER during mitotic exit. Further, they suggest that advanced dephosphorylation of CHMP7 in a chromatin associated pool licenses its polymerisation, ensuring that the nucleus reforms with, and retains, its proper complement of INM proteins for the following interphase.

## Discussion

While ESCRT-III is necessary for NE regeneration, during M-phase the cell is challenged by the problem of how to suppress ESCRT-III assembly when regulated nucleocytoplasmic transport mechanisms are inactive and both CHMP7 and its activator of polymerisation, LEM2, are in the same hybrid ER/

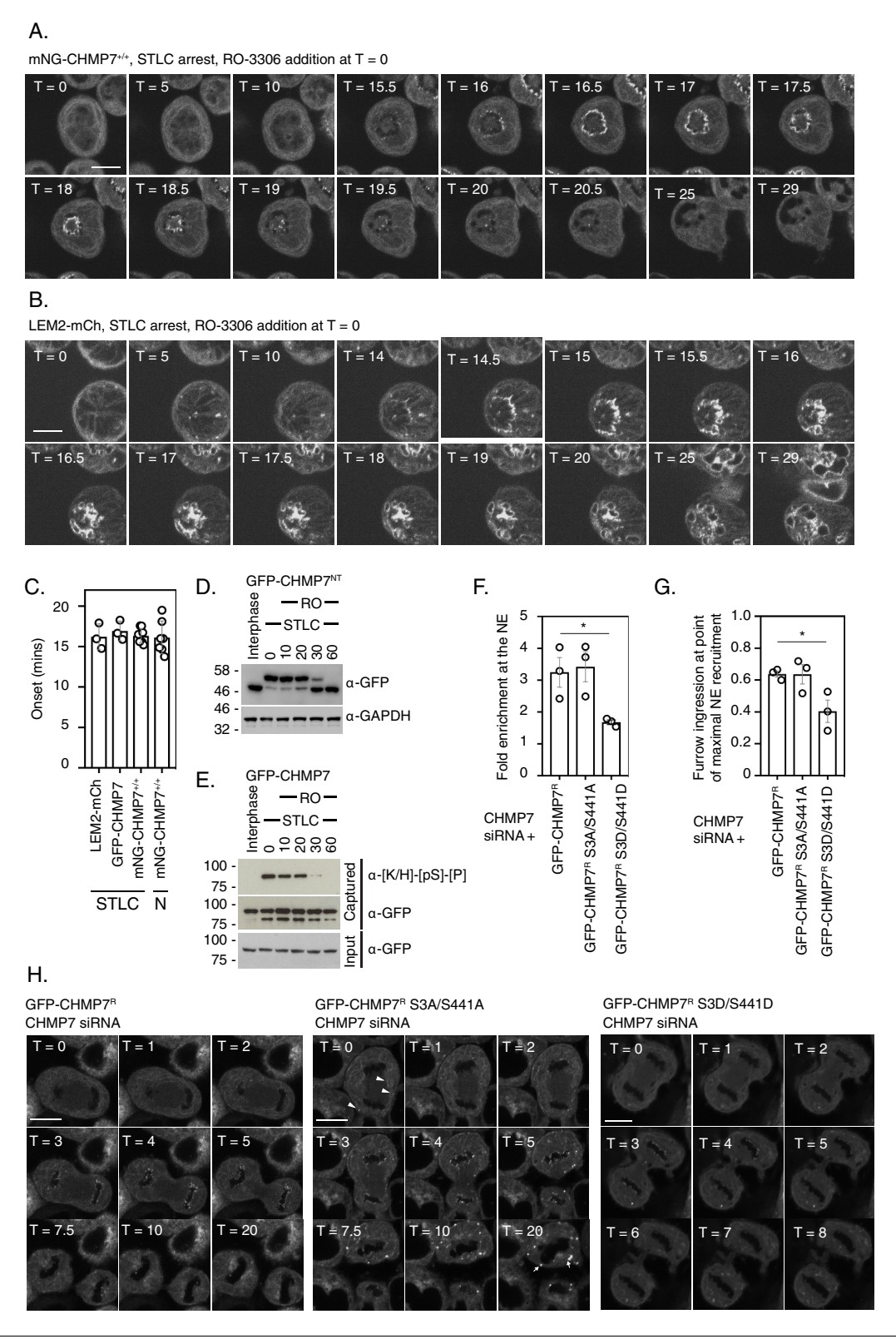

**Figure 5.** Dephosphorylation kinetics of CHMP7 during mitotic exit. (**A, B**) CAL-51 cells homozygously edited to express mNG-CHMP7 (**A**) or HeLa cells stably expressing LEM2-mCh (**B**) were synchronised to prometaphase with STLC and then released through addition of RO-3306. Cells were imaged live through synchronised M-exit. (**C**) Quantification of assembly onset (assembly duration presented in *Figure 5—figure supplement 1D*) post RO3306 release. mNG-CHMP7$^{+/+}$ (onset 16.3 ± 0.88 mins; duration 3.1 ± 0.44 mins; N = 9, n = 104) or LEM2-mCh (onset 16.2 ± 2.22 mins; duration 4.1 ± 0.44

*Figure 5 continued on next page*

*Figure 5 continued*

mins; N = 3, n = 80). Additional quantification of the above parameters from HeLa cells stably expressing GFP-CHMP7 (onset 16.9 ± 1.21 min; duration 3.3 ± 0.13 min; N = 3, n = 97) and subject to STLC arrest and RO3306 release (from *Figure 4—figure supplement 1*) and CAL-51 cells homozygously edited to express mNG-CHMP7 (onset 16.1 ± 2.02 min; duration 4.2 ± 2.19 min; N = 7, n = 39) and subject to nocodazole arrest and RO3306 release (from *Figure 5—figure supplement 1*) are also presented in C. A one-way ANOVA with Tukey's multiple comparisons revealed no significant differences between the datasets. (D). (E) HeLa cells stably expressing GFP-CHMP7^NT (D) or GFP-CHMP7 (E) were either untreated or arrested in mitosis through STLC inhibition and forced out of mitosis using RO3306 for the indicated times. Lysates in D, reporting Ser3 dephosphorylation were resolved by PhosTag or normal SDS-PAGE and examined with antisera raised against GFP or GAPDH respectively. Lysates from E, reporting Ser441 dephosphorylation were immunoprecipitated using GFP-trap resin and both inputs and captured fractions were examined by western blotting with antisera raised against phosphorylated CDK1 substrates and GFP (N = 3). (F-H) CHMP7-depleted HeLa cells stably expressing GFP-CHMP7^R, GFP-CHMP7^R S3A/S441A, or GFP-CHMP7^R S3D/S441D were imaged live during mitotic exit (H). In F, the fold-enrichment of GFP-signal at the reforming nuclear envelope was calculated from three independent experiments, presented as mean (N = 3) ± S.E.M. (WT, n = 24; S3A/S441A, n = 27, n.s. (p = 0.92), S3D/S441D, n = 52, p = 0.049; one-way ANOVA with Dunnett's multiple comparisons). In G, the degree of furrow ingression (midzone diameter/daughter cell diameter) at the point of maximal GFP-CHMP7 recruitment was used as a proxy of progression through M-exit and was calculated from three independent experiments, presented as mean (N = 3) ± S.E.M. (WT, n = 24; S3A/S441A, n = 24, n.s. (p = 0.99), S3D/S441D, n = 46, p = 0.042; one-way ANOVA with Dunnett's multiple comparisons). In H, for cells stably expressing GFP-CHMP7^R S3A/S441A, arrowheads depict precocious clustering of CHMP7 in the peripheral ER during anaphase that develop into larger clusters during mitotic exit. In all panels, time in minutes and a scale bar of 10 μm are reported.

The online version of this article includes the following video, source data, and figure supplement(s) for figure 5:

**Source data 1.** Dynamics of CHMP7 assembly at the reforming nuclear envelope.
**Figure supplement 1.** Analysis of the kinetics of GFP-CHMP7 and LEM2-mCh assembly at the reforming NE during monopolar mitotic exit.
**Figure supplement 1—source data 1.** Kinetics of CHMP7 and LEM2 assembly, and LEM2 cluster dissolution, at the reforming monopolar NE.
**Figure supplement 2.** During M-exit, the nuclear-associated pool of CHMP7 is dephosphorylated in advance of the cytosolic/ER-associated pool.
**Figure supplement 2—source data 1.** Kinetics of CHMP7 dephopshorylation in chromatin-associated and extra-chromatin associated fractions during M-exit and siRNA sensitivity of the pSer3 CHMP7 antisera.
**Figure supplement 3.** CDK phosphorylation on CHMP7 Ser3 and Ser441 suppresses CHMP7 assembly and inappropriate capture of ESCRT-III subunits during mitotic exit.
**Figure supplement 3—source data 1.** CDK1 phosphorylation of CHMP7 suppresses formation of clusters of CHMP7 that grow during M-exit.
**Figure 5—video 1.** Related to *Figure 5A*.
https://elifesciences.org/articles/59999#fig5video1
**Figure 5—video 2.** Related to *Figure 5—figure supplement 1A*.
https://elifesciences.org/articles/59999#fig5video2
**Figure 5—video 3.** Related to *Figure 5A*.
https://elifesciences.org/articles/59999#fig5video3
**Figure 5—video 4.** Related to *Figure 5—figure supplement 1B*.
https://elifesciences.org/articles/59999#fig5video4
**Figure 5—video 5.** Related to *Figure 5—figure supplement 1C*.
https://elifesciences.org/articles/59999#fig5video5
**Figure 5—video 6.** Related to *Figure 5—figure supplement 1E*.
https://elifesciences.org/articles/59999#fig5video6
**Figure 5—video 7.** Related to *Figure 5H*.
https://elifesciences.org/articles/59999#fig5video7

NE membrane. During M-exit, the cell must also convert an apolar sheet of ER/NE membrane to a polarised sheet with distinct INM and ONM identities. We found that during mitotic exit, the INM protein LEM2 was co-enriched with CHMP7 at the reforming nuclear envelope and then released in a CHMP7-dependent manner to populate the INM. Importantly, LEM2 enrichment at the reforming nuclear envelope occurred in the absence of CHMP7, consistent with it being an upstream regulator of this pathway. The LEM2 clusters in CHMP7-depleted cells that persisted into the next interphase led to a breakdown of nucleocytoplasmic compartmentalisation and the generation of localised DNA damage. By inducing precocious assembly of CHMP7 through deletion of a C-terminal autoregulatory helix, we discovered that clusters of CHMP7^δHelix6 that formed were disassembled upon mitotic entry, suggesting that CHMP7 assembly may be regulated by cell cycle control mechanisms. Mitotic phosphorylation of the ESCRT-III components IST1 and CHMP4C is already known to regulate cytokinetic abscission (*Carlton et al., 2012*; *Capalbo et al., 2012*; *Caballe et al., 2015*), suggesting that this machinery is capable of being modified by phosphorylation, and mitotic phosphorylation of LEM2 can control both its interaction with partner proteins such as Barrier to

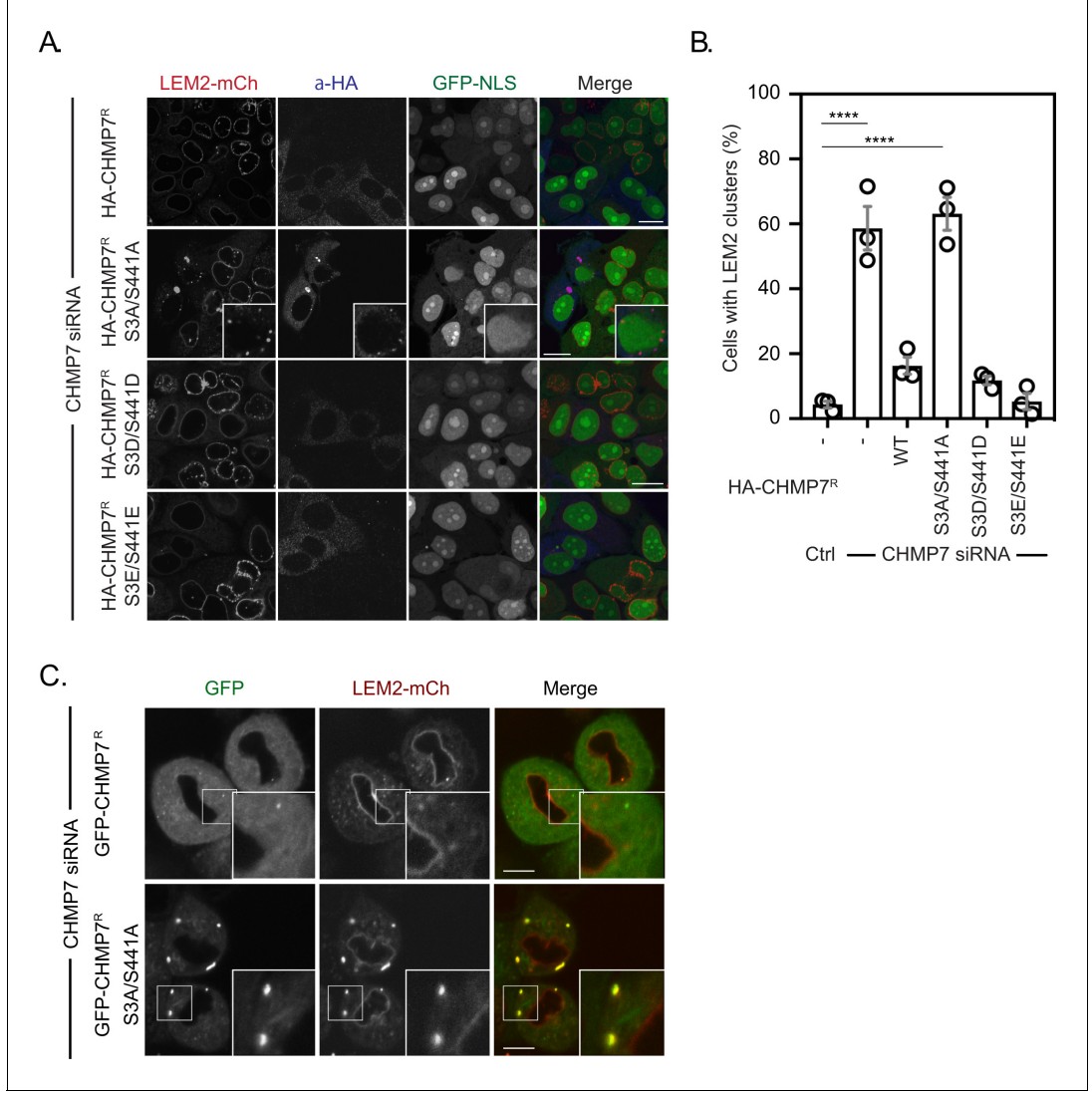

**Figure 6.** Mitotic CHMP7 phosphorylation suppresses inappropriate clustering of LEM2 during M-exit. (**A**) HeLa cells stably expressing LEM2-mCh and GFP-NLS were treated with CHMP7-targetting siRNA and transfected with plasmids encoding the indicated HA-CHMP7$^R$ constructs, fixed and stained with antisera raised against HA. (**B**) The number of cells displaying extra-NE clusters of LEM2-mCh was quantified from A, presented as mean ± S.E.M from N = three independent experiments and the significance of LEM2 cluster formation was assessed by one-way ANOVA with Dunnett's post-hoc test (Control, n = 416; CHMP7 siRNA, n = 916, p < 0.0001; CHMP7 siRNA + HA-CHMP7$^R$ n = 375, n.s., p = 0.159; CHMP7 siRNA + HA-CHMP7$^R$ S3A/S441A, n = 379, P < 0.001; CHMP7 siRNA + HA-CHMP7$^R$ S3D/S441D, n = 192, n.s., p = 0.529; CHMP7 siRNA + HA-CHMP7$^R$ S3E/S441E, n = 251, n.s., p = 0.999). (**C**) HeLa cells stably expressing LEM2-mCh and either GFP-CHMP7$^R$ or GFP-CHMP7$^R$ S3A/S441A were transfected with siRNA targeting CHMP7 and imaged during mitotic exit. Precocious clusters were observed in 15/49 (WT) and 12/15 (S3A/S441A) imaged cells. In all panels, time in minutes and a scale bar of 10 μm is depicted.

The online version of this article includes the following video, source data, and figure supplement(s) for figure 6:

**Source data 1.** CHMP7 phosphorylation prevents inappropriate LEM2 clusters forming in the peripheral ER during M-exit.

**Figure supplement 1.** Failure to phosphorylated CHMP7 Ser3 and Ser441 during M-phase results in inappropriate clustering and capture of LEM2 in the peripheral ER that compromises its localisation to the INM.

**Figure supplement 1—source data 1.** Nuclear envelope compartmenatlisation in the presence of phosphomutant or phosphomutant versions of CHMP7.

**Figure 6—video 1.** Related to *Figure 6C*.
https://elifesciences.org/articles/59999#fig6video1

**Figure 6—video 2.** Related to *Figure 6—figure supplement 1C and D*.
https://elifesciences.org/articles/59999#fig6video2

Autointegration Factor (BAF) (*Asencio et al., 2012*) and its phase-separation properties (*von Appen et al., 2020*). We identified direct CDK1 phosphorylation of CHMP7 at Ser3 and Ser441 as a suppressor of both CHMP7's ability to interact with LEM2 and its ability to assemble, as judged by sedimentation assays. Given the requirement for LEM2 in the activation of CHMP7 polymerisation, introducing phosphomimetic residues at these positions suppressed LEM2-dependent CHMP7 polymerisation in vitro. Interestingly, CHMP7 is the only ESCRT-III subunit to bear recognisable consensus sequences for CDK1 phosphorylation, suggesting that mitotic regulation of ESCRT-III may be effected through this protein. Analysing the kinetics of CHMP7 phosphorylation revealed that phosphorylation coincided with M-phase entry and that global levels of CHMP7 phosphorylation persisted during the timeframe of ESCRT-III assembly at the reforming NE. We found that a local pool of CHMP7 at the reforming NE was dephosphorylated in advance of the major pool of CHMP7 in the cytosol and on the peripheral ER, suggesting a mechanism by which spatiotemporal regulation of ESCRT-III assembly could be controlled locally by CHMP7 phosphorylation and dephosphorylation. Whilst phosphomimetic residues at the CDK1-phosphorylation sites suppressed CHMP7's interaction with LEM2 and reduced its sedimentation in a minimal in vitro system, their effect in cells were more complex to interpret, as we found that CHMP7 bearing phosphomimetic mutations could still be recruited, albeit with a reduced intensity and in a delayed manner, to the reforming NE, and could still rescue the phenotypes of CHMP7-depletion. It is possible that the restriction on assembly imposed by phosphomimetic residues is purposefully weak, allowing it to be overridden by the high concentration of LEM2 at the reforming NE. Alternatively, there may be additional LEM2-independent activators of CHMP7 polymerisation in cells. Examination of nonphosphorylatable substitutions were clearer to interpret as these versions of CHMP7 led to correct nuclear envelope assembly, but additional spatially inappropriate capture of residual LEM2 in the peripheral ER during M-exit. These clusters persisted into interphase and captured progressively more LEM2, leading to the loss of LEM2 from the INM. Whilst CDK1 phosphorylation at these residues suppresses inappropriate CHMP7 clustering and LEM2 capture during M-exit, it remains unclear why these clusters develop only during telophase, rather than earlier phases of mitosis. Given that LEM2 similarly undergoes M-phase phosphorylation (*von Appen et al., 2020*) and likely dephosphorylation during M-exit, we speculate its dephosphorylation kinetics may contribute to its co-association with CHMP7. Finally, contrary to the prevailing view that this assembly occurs at sites where spindle microtubules traverse the reforming nuclear envelope, we found that microtubules are dispensable for CHMP7 assembly at this organelle. These data are consistent with the findings from ESCRT-dependent interphase nuclear envelope repair where no such trans-nuclear envelope microtubules exist (*Raab et al., 2016*; *Denais et al., 2016*; *Robijns et al., 2016*). These data link ESCRT-III assembly during mitotic exit to classical cell cycle control mechanisms and identify mitotic phosphorylation of CHMP7 as a mechanism to directly suppress its inappropriate assembly allowing LEM2 to correctly populate the INM.

## Materials and methods

### Key resources table

| Reagent type (species) or resource | Designation | Source or reference | Identifiers | Additional information |
|---|---|---|---|---|
| Gene (*Homo sapiens*) | CHMP7 | Ensembl | ENSG00000147457 | |
| Gene (*Homo sapiens*) | LEMD2 | Ensembl | ENSG00000161904 | Also called LEM2 |
| Strain, strain background (*Escherichia coli*) | NEB5a | NEB | C2987 | Cloning grade derivative of *E. coli* DH5a |
| Strain, strain background (*Escherichia coli*) | BL21Star (DE3) | Invitrogen | C601003 | Protein production strain of *E. coli* DH5a |
| Cell line (*Homo sapiens*) | HeLa | ATCC | | STR profiled from Crick Cell Services |

*Continued on next page*

Continued

| Reagent type (species) or resource | Designation | Source or reference | Identifiers | Additional information |
|---|---|---|---|---|
| Cell line (*Homo sapiens*) | CAL-51 | PMID:27618263 | | STR profiled from Crick Cell Services |
| Cell line (*Homo sapiens*) | 293T | ATCC | | STR profiled from Crick Cell Services |
| Cell line (*Homo sapiens*) | GP2-293 | Clontech | | STR profiled from Crick Cell Services |
| Cell line (*Homo sapiens*) | CAL-51 mNG-CHMP7$^{+/+}$ | PMID:27618263 | | STR profiled from Crick Cell Services |
| Cell line (*Homo sapiens*) | HeLa GFP-CHMP7$^R$ | PMID:27618263 | | STR profiled from Crick Cell Services |
| Transfected construct (*Homo sapiens*) | pCMS28-EcoRI-NotI-XhoI | PMID:17556548 | | Bicistronic retroviral packaging and exprssion vector expressing IRES-PuroR downstream of the MCS. GFP-fusions of CHMP7 were constructed in this vector |
| Transfected construct (*Homo sapiens*) | pCMS28-EcoRI-NotI-XhoI-L-GFP | This study | | A version of pCMS28 containing a LAP tag, based upon PMID:15644491 |
| Transfected construct (*Homo sapiens*) | pMSCVneo EcoRI-NotI-XhoI | This study | | A kind gift from Prof Martin-Serrano (King's College London) Modified version of Clontech's pMSCV-neo. LEM2-mCherry fusions were expressed from this vector. A retroviral packaging vector that encodes neomycin resistance from a second promoter. |
| Transfected construct (*Homo sapiens*) | pLHCX-MCS-HA-CHMP7$^R$ | This study | | Retroviral packaging and expression vector (a kind gift from Prof Tony Ng, King's College London) into which HA-CHMP7$^R$ was inserted and mutagenised as necessary. |
| Transfected construct (*Homo sapiens*) | pCR3.1 YFP/HA EcoRI-NotI-XhoI | PMID:14519844 | | Mammalian expression vector for YFP- or HA-fusion proteins |
| Transfected construct (*Homo sapiens*) | pCAGGS/GST | PMID:14519844 | | Mammalian expression vector for GST-fusions proteins |
| Transfected construct (*Homo sapiens*) | pCMVdR8.91 | A kind gift from Dr Caroline Goujon (IRIM CNRS, Montpellier) | | Packaging plasmid used for lentivirus production. Mammalian expression of Gag-Pol driven by CMV promoter. |
| Transfected construct (*E. coli*) | pET28a-NusA-EcoRI-NotI-XhoI | This paper | | A version of pET28a comprising N-terminal tandem His$_6$ affinity tag, an N-utilisation sequence A (NusA) solubilisation tag. |
| Antibody | GAPDH (Mouse monoclonal) | Merck Millipore | Clone 6C5, MAB374 | (1:10,000) |
| Antibody | Calnexin (Rabbit polyclonal) | Abcam | ab22595 | (1:1000) |
| Antibody | Alpha-Tubulin (Mouse monoclonal) | Merck Millipore | Clone DM1A, CP06 | (1:500) |
| Antibody | CHMP2A (Rabbit polyclonal) | Proteintech | 10477–1-AP | (1:1000) |
| Antibody | IST1 (Rabbit polyclonal) | Proteintech | 51002–1-AP | (1:1000) |
| Antibody | CHMP7 (Rabbit polyclonal) | Proteintech | 16424–1-AP | (1:1000) |

*Continued on next page*

*Continued*

| Reagent type (species) or resource | Designation | Source or reference | Identifiers | Additional information |
|---|---|---|---|---|
| Antibody | LAP1 (Rabbit polyclonal) | Proteintech | 21459–1-AP | (1:1000) |
| Antibody | Emerin (Rabbit polyclonal) | Proteintech | 16424–1-AP | (1:1000) |
| Antibody | GFP (Mouse monoclonal) | Roche | Clone 7.1/13.1, 11814460001 | (1:1000) |
| Antibody | mCherry (Rabbit polyclonal) | Abcam | ab167453 | (1:1000) |
| Antibody | HA.11 (Mouse monoclonal) | Biolegend | Clone 16B12, 901502 | (1:1000) |
| Antibody | gH2AX (Mouse monoclonal) | Merck Millipore | Clone JBW301, 05–636 | (1:200) |
| Antibody | 53BP1 (Rabbit polyclonal) | Novus Biologicals | NB100-305 | (1:200) |
| Antibody | Phospho-CDK Substrate Motif [(K/H)pSP] (Rabbit monoclonal) | Cell Signaling Technology | 9477 | (1:1000) |
| Antibody | Phospho-MAPK/CDK Substrates [PXS*P or S*PXR/K] (Rabbit monoclonal) | Cell Signaling Technology | Clone 34B2, 2325 | (1:1000) |
| Antibody | Phospho-Histone H3 (Ser10) Antibody (Rabbit polyclonal) | Cell Signaling Technology | 9701 | (1:1000) |
| Antibody | LEM2 (Rabbit polyclonal) | Merck Millipore | HPA017340 | (1:1000) |
| Antibody | HSP 90α/β Antibody (Mouse monoclonal) | Santa Cruz Biotechnology | Clone F8, sc-13119 | (1:1000) |
| Antibody | CHMP7 pSer3 | This study | 3891 | (1:500) In-house produced rabbit polyclonal phospho S3 CHMP7 antisera. |
| Antibody | CHMP7 pSer3 | This study | 3892 | (1:500) In-house produced rabbit polyclonal phospho S3 CHMP7 antisera. |
| Chemical compound, drug | PhosTag Acrylamide | Fujifilm Wako Chemicals | AAL-107 | |
| Chemical compound, drug | RO-3306 | Merck Millipore | SML-0569 | |
| Chemical compound, drug | S-Trityl L-Cysteine | Merck Millipore | 164739 | |
| Chemical compound, drug | Nocodazole | Merck Millipore | M1404 | |
| Peptide, recombinant protein | CDK1:CyclinB1 complex | Merck Millipore | 14–450M-D | |
| Chemical compound, drug | HALT Protease Inhibitors | Fisher Scientific | 78429 | |
| Chemical compound, drug | Complete EDTA-free protease inhibitors | Merck Millipore | 11836170001 | |
| Chemical compound, drug | PhosSTOP phosphatase inhibitor tablets | Merck Millipore | PHOSS-RO | |
| Commercial assay, kit | GFP-Trap Magnetic Agarose | Chromotek | Gtma-20 | |

*Continued on next page*

*Continued*

| Reagent type (species) or resource | Designation | Source or reference | Identifiers | Additional information |
|---|---|---|---|---|
| Commercial assay, kit | Glutathione Sepharose 4B GST-tagged protein purification resin | GE Healthcare | 17075601 | |
| Commercial assay, kit | Dynabeads antibody coupling kit | Invitrogen | 14311D | |
| Sequence-based reagent | Control siRNA | Horizon Discovery | D-001810 | |
| Sequence-based reagent | CHMP7 siRNA-1 | Horizon Discovery, PMID:27618263 | GGGAGAAGA TTGTGAAGTTdTdT | |
| Sequence-based reagent | CHMP7 siRNA-1 | Horizon Discovery, PMID:27618263 | GGAGGUGU AUCGUCUGUAUdTdT | |
| Sequence-based reagent | IST-1 siRNA | Horizon Discovery | M-020977 | |
| Sequence-based reagent | LEM2 siRNA | Horizon Discovery | M-017941 | |
| Other | | | | |

## Cell culture

STR-profiled, mycoplasma-free vials of HeLa, 293T, GP2-293 and CAL-51, CAL-51 mNG-CHMP7$^{+/+}$ or HeLa cells stably expressing GFP-CHMP7$^R$ have been described previously (*Olmos et al., 2016*) and were banked and obtained from the Crick Cell Services Science Technology Platform. Cells were cultured in DMEM containing 10% FBS, Penicillin (100 U/mL) and Streptomycin (0.1 mg/mL). Stable cells lines were generated by transduction using MLV-based retroviruses or HIV-1-based lentiviruses as described previously (*Olmos et al., 2015*), and selected using Puromycin (1 µg/mL) or G418 (750 µg/mL) as necessary. Where necessary, cells were sorted to monoclonality by limiting dilution or FACS.

## Plasmids

Retroviral expression vectors containing the human CHMP7 coding sequence (both WT, siRNA-resistant and N-terminal scanning mutagenesis) have been described previously (*Olmos et al., 2016*). A codon optimised human LEM2 sequence was obtained by genesynthesis (GeneWIZ) and cloned *EcoRI-NotI* into a modified version of pMSCVneo bearing an *EcoRI-NotI-XhoI* polylinker leading into mCherry. Derivatives and mutations of CHMP7 and LEM2 were obtained by standard molecular biology protocols and cloned into pCMS28-GFP-CHMP7, pLHCX-HA-CHMP7$^R$, or pMSCVneo LEM2-mCh as required. A cDNA for VPS4A was cloned with a C-terminal LAP tag (*Cheeseman and Desai, 2005*) into pCMS28. A lentiviral expression vector for GFP-NLS was prepared by adding the c-MYC NLS to the C-terminus of GFP in pLVXP-GFP (a kind gift from Dr Michael Way, Crick). The VPS4 MIT domain (residues 1–122) was amplified by PCR and cloned *EcoRI-NotI* sites of pCAGGS-GST *EcoRI-NotI-XhoI*. To replace the putative MIM in CHMP7 Helix6 with a known MIM from CHMP2A, CHMP7 δHelix6 was prepared by exchanging residues 420–428 with a *BamHI* restriction site. Oligos encoding the CHMP2A MIM (reverse translation of Helix6: SALADADADLEERLKNLRRD) were cloned into the *BamHI* site. For expression of recombinant proteins, the open reading frames of full-length CHMP7 and LEM2 C-terminal domain 395–503 (LEM2$^{CT}$), or derivatives, were cloned with an N-terminal TEV-recognition site that cleaves at the 1$^{st}$ Methionine as *EcoRI-NotI* inserts into a version of pET28a comprising N-terminal tandem His$_6$ affinity tag, an N-utilisation sequence A (NusA) solubilisation tag (*Davis et al., 1999*) and *EcoRI-NotI-XhoI* polylinker.

## Generation of stable cell lines

For retroviral transduction, above constructs in retroviral packaging vectors were transfected with pVSVG into GP2-293 cells (Clontech), constructs in lentiviral packaging vectors were transfected with pCMV8.91 and pVSVG into 293 T cells. Supernatants were harvested, clarified by centrifugation (200

x g, 5 min), filtered (0.45 μm) and used to infect target cells in the presence of 8 μg/mL polybrene (Millipore) at MOI < 1. Antibiotic selection was applied after 48 hr.

## Antibodies

An antibody against GAPDH (MAB374) was from Millipore; Calnexin (ab22595) was from Abcam; Tubulin (DM1A) was from Sigma; CHMP2A (10477–1-AP) was from Proteintech; IST1 (51002–1-AP) was from Proteintech; CHMP7 (16424–1-AP) was from Proteintech; GFP (7.1/13.1) was from Roche; mCherry (ab167453) was from Abcam; HA.11 (16B12) was from (Biolegend); γH2AX (05–636) was from Sigma; 53BP1 (NB100-305) was from (Novus Biologicals). Anti-CDK1 substrate antibodies were from Cell Signaling Technology (9477S and 2325S). Anti-Histone H3 pS10 was from Cell Signaling Technology (9701S). Anti-LEM2 was from Sigma (HPA017340); anti-HSP90 (F8) was from Santa Cruz; anti-Emerin (10351–1-AP) was from Proteintech. Anti-LAP1 (21459–1-AP) was from Proteintech. Alexa conjugated secondary antibodies were from Invitrogen and HRP-conjugated secondary antibodies were from Millipore. IRDye 800 CW (925–32210) and IRDye 680 RD (925–68071) were from LI-COR Biosciences. Anti-peptide antibodies against CHMP7 pSer3 (3891 and 3892) were generated by immunisation of rabbits with KLH-conjugated peptides (MWpSPEREAEAPAGGC) by GenScript Biotech (Netherlands) B.V.

## SDS-PAGE and western blotting

Cell lysates and fractions were denatured by boiling in denaturing LDS-sample buffer (Life Technologies) and resolved using SDS-PAGE using precast Novex gels (Life Technologies). Resolved proteins were transferred onto nitrocellulose by western blotting (wet transfer, 100V, 1 hr) and were probed with the indicated antisera in 5% milk or BSA. HRP-conjugated secondary antibodies were used to probe membranes, following by incubation with ECL Prime enhanced chemiluminescent substrate (GE Healthcare) and visualised by exposure to autoradiography film; alternatively, IRdye fluorophore conjugated secondary antibodies were used to probe membranes, following by visualisation by infra-red imaging (LICOR Odyssey Fc). Where necessary, 10% polyacrylamide gels were prepared manually containing 200 μM $MnCl_2$ and either 25 μM PhosTag-acrylamide (whole cell lysates) or 50 μM PhosTag-acrylamide (immunoprecipitants). PhosTag-acrylamide was from Fujifilm Wako Chemicals. EDTA-free Laemmli buffer was used to prepare lysates for PhosTag-gels, prestained molecular weight markers (NEB) were supplemented with twice the volume of 10 mM $MnCl_2$ and 1 vol of 4x EDTA-free Laemmli buffer. After electrophoresis, PhosTag gels were soaked for 10 min in Tris-Gly transfer buffer containing 10 mM EDTA, 10 min in Tris-Gly transfer buffer containing 1 mM EDTA and 10 min in Tris-Gly transfer buffer prior to electro-transfer.

## Transient transfection of cDNA

HeLa and CAL-51 cells were transfected using Lipofectamine-3000 (Life Technologies) according to the manufacturer's instructions. 293GP2 cells were transfected using linear 25 kDa polyethylenimine (PEI, Polysciences, Inc), as described previously (*Carlton and Martin-Serrano, 2007*).

## siRNA transfections

HeLa cells were seeded at a density of 1E5 cells/mL and were transfected with siRNA at 20 nM, 2 hr after plating using RNAi-MAX (Invitrogen), for 72 hr. The following targeting sequences that have previously been demonstrated to achieve potent and specific suppression of the targeted CHMP were employed: Control, Dharmacon Non-targeting control D-001810; CHMP7-1 (GGGAGAAGA TTGTGAAGTTdTdT), CHMP7-2 (GGAGGUGUAUCGUCUGUAUdTdT); LEM2 and IST1, siGenome Dharmacon SmartPools.

## Recombinant proteins

Both CHMP7 and LEM2$^{CT}$ were purified as previously described (*Gu et al., 2017*; *von Appen et al., 2020*) with some modifications. *Escherichia coli* BL21(DE3) cells expressing the 2xHis$_6$-NusA-TEV-tagged proteins were grown at 37°C to OD 0.6 in LB medium, transferred to 18°C and induced with 0.25 mM IPTG overnight. Cells expressing full length CHMP7, CHMP7 S3D/S441D, or CHMP7 S3E/D441E were harvested by centrifugation and lysed with TNG1000 lysis buffer (50 mM Tris-HCl pH 7.5, 1 M NaCl, 10% w/v Glycerol) supplemented with 1 tablet of cOmplete Protease Inhibitors

(Roche), 20 mM imidazole, 5 mM β-Mercaptoethanol, 0.2% Triton X-100, 1 mM PMSF, 10 µg/mL DNaseI, and 1 mg/mL lysozyme. Cells were then sonicated and clarified by centrifugation at 40,000 $\times$ g for 60 min. Supernatants were incubated with Ni-NTA beads at 4°C for 2 hr. Beads were extensively washed with complete TNG1000 lysis buffer, transferred into a TNG1000 buffer with 0.01% Triton X-100, and eluted in four steps of this buffer containing stepwise increasing imidazole concentrations (50, 150, 250, and 400 mM). The eluted protein was dialysed overnight against TNT buffer (50 mM Tris-HCl pH 7.6, 150 mM NaCl, 0.5 mM TCEP) with 5% glycerol in the presence of recombinant TEV protease. The sample was further dialysed for two more times in fresh TNT buffer, before incubating with new Ni-NTA beads for re-capturing the 2xHis$_6$-NusA tag. The supernatant was then injected into a Superdex 200 Increase 10/300 GL column pre-equilibrated with TNT buffer, monomeric CHMP7 fractions were pulled, aliquoted and snap-frozen in liquid nitrogen. 4 L of bacterial culture yielded 1 mg of CHMP7 wildtype, 1.8 mg of CHMP7 S3D/S441D and 1.6 mg of CHMP7 S3E/S441E. BL21(DE3) cells expressing LEM2$^{CT}$ or HA-LEM2$^{CT}$ were processed similarly, with some modifications: lysis and washes were carried out in TNG500 lysis buffer (50 mM Tris-HCl pH 7.5, 500 mM NaCl, 10% w/v Glycerol); no stepwise imidazole elution was performed; TEV cleavage was achieved on the Ni-bound protein overnight. Fractions containing LEM2$^{CT}$ were pulled, concentrated, aliquoted and snap-frozen in liquid nitrogen. Yield was 4 mg of protein per 3 L of bacterial culture. Before subsequent experiments, proteins were thawed quickly and cleared by centrifugation at 22,000 $\times$ g for 20 min.

### In vitro kinase assay

10 µM of recombinant CHMP7 was incubated in TNT buffer (50 mM Tris-HCl pH 7.6, 150 mM NaCl, 0.5 mM TCEP) with 10 mM MgCl$_2$, 1 µg of recombinant CDK1:CyclinB1 active complex (Millipore) in the presence or absence of 3 mM ATP. After 60 min at 30°C, samples were spun in a benchtop centrifuge at 22,000 $\times$ g for 40 min at 4°C. Supernatant (S) and Pelleted (P) fractions were resuspended in equal volumes of 1x Laemmli buffer, run on a PhosTag SDS-PAGE gel. Samples and gels were prepared as described above. Western-blot band intensities were quantified by densitometry using ImageJ and ratios of insoluble pelleted proteins were calculated.

### CHMP7 sedimentation assay

10 µM of recombinant CHMP7, CHMP7 S3D/S441D, or CHMP7 S3E/S441E was incubated in TNT buffer (50 mM Tris-HCl pH 7.6, 150 mM NaCl, 0.5 mM TCEP) for 60 min at 30°C. Samples were spun in a benchtop centrifuge at 22,000 $\times$ g for 40 min at 4°C. Supernatant (S) and Pelleted (P) fractions were resuspended in equal volumes of 1x Laemmli buffer, run on an SDS-PAGE gel and analysed by western blotting with antisera raised against CHMP7. Band intensities were quantified by densitometry using ImageJ and ratios of insoluble pelleted proteins were calculated.

### CHMP7-LEM2 sedimentation assay

5 µM CHMP7, CHMP7 S3D/S441D or CHMP7 S3E/S441E was combined with 10 µM HA-LEM2$^{CT}$ and diluted in a final volume of 50 µL TKD buffer (50 mM Tris HCl pH 8, 150 mM KCl, 1 mM DTT) supplemented with 1 x cOmplete EDTA free protease inhibitors and incubated at room temperature for 2 hr. Samples were spun in a benchtop centrifuge at 22,000 $\times$ g for 40 min at 4°C. Supernatant (S) and Pelleted (P) fractions were resuspended in equal volumes of 1x Laemmli buffer, run on a SDS-PAGE gel and detected by western blot using antisera raised against CHMP7 or HA-tag. Band intensities were quantified by densitometry using ImageJ and ratios of insoluble pelleted proteins were calculated.

### CHMP7-LEM2 interaction assay

Anti-CHMP7 (Proteintech, 16424–1-AP) was covalently coupled to Dynabeads (Life Technologies) at a final concentration of 10 mg/mL, according to the manufacturer's instructions. Either CHMP7 or CHMP7 S3D/S441D was combined with HA-LEM2$^{CT}$ at a ratio of 1:2 (µM: µM), diluted in a final volume of 100 µL TKD buffer (50 mM Tris HCl pH 8, 150 mM KCl, 1 mM DTT) supplemented with 1 x cOmplete EDTA-free protease inhibitors and incubated overnight with rotation at 4°C. 5 µg anti-CHMP7 coupled Dynabeads per reaction were washed in TKD, diluted in 100 µl TKD containing 0.1% Tween-20 and incubated with the interacting proteins for 10 min with rotation at 4°C. Immune-

complexes were isolated by magnetisation and washing thrice with TKD containing 0.05% Tween-20. Inputs were collected before Dynabead-addition, denatured input and captured fractions were analysed by SDS-PAGE.

## CDK1-phosphorylated CHMP7-LEM2 interaction assay

10 µM of recombinant CHMP7 was incubated in TNT buffer (50 mM Tris-HCl pH 7.6, 150 mM NaCl, 0.5 mM TCEP) with 10 mM $MgCl_2$, 1 µg of recombinant CDK1:CyclinB1 active complex (Millipore) in the presence or absence of 3 mM ATP. After 60 min at 30°C, the reaction was combined with 10 µM HA-LEM2$^{CT}$ to a final CHMP7:LEM2$^{CT}$ ratio of 1:2 (5 µM:10 µM), diluted in a final volume of 100 µL HiSalt buffer (50 mM Tris-HCl pH 8, 250 mM KCl, 1 mM DTT) and dialysed overnight on Dial-A-Lyzer Mini 3,500 MWCO cassettes (Thermo Scientific) at 4°C into LoSalt buffer (30 mM HEPES pH 8, 25 mM KCl, 1 mM DTT)[15]. Immune-complexes were isolated by magnetisation with anti-CHMP7 coupled Dynabeads as described above.

## Negative stain electron microscopy

CHMP7 or CHMP7 S3D/S441D was combined with HA-LEM2$^{CT}$ at a ratio of 1:2 (3 µM:6 µM), diluted in a final volume of 50 µL HiSalt buffer (50 mM Tris-HCl pH 8, 250 mM KCl, 1 mM DTT) and dialysed overnight on Dial-A-Lyzer Mini 3,500 MWCO cassettes (Thermo Scientific) at 4°C into LoSalt buffer (30 mM HEPES pH 8, 25 mM KCl, 1 mM DTT)[15]. A thin carbon film was deposited on a freshly cleaved mica sheet in a carbon evaporator (Emitech K950x, 100 mA, four pulses), then transferred onto TEM grids (400 mesh copper, TAAB). The grids were then glow discharged (Emitech K100X) at 25 mA for 30 s in air. A 4 µL sample solution was pipetted onto the glow-discharged carbon-filmed TEM grids. The solution was then blotted from the side of the grid with filter paper after 2 min, followed by staining with 1% Sodium Silicotungstate (SST) aqueous solution for 30 s. Data were collected with a CCD camera in a Tecnai Spirit TEM (FEI) operated at 120 kV.

## Fixed cell imaging

HeLa cells were fixed in MeOH or 4% PFA and subject to processing for immunofluorescence as described previously (*Carlton and Martin-Serrano, 2007*). Cells were imaged using a Nikon Eclipse Ti2 microscope teamed with an Andor Dragonfly 200 Spinning Disc Confocal imaging system and paired with Andor Zyla sCMOS and Andor iXon EM-CCD cameras. Images were processed in FIJI and exported to Photoshop for assembly into figures. For rescue of LEM2 clustering (*Figure 5*), cells stably expressing LEM2-mCh and GFP-NLS were transfected in 8-well Ibidi chamber slides with control or CHMP7-targetting siRNA as described above. Media was changed after 6 hr. After 16 hr, cells were transfected with 100 ng pLHCX-CHMP7$^R$ plasmids using Lipofectamine 3000. Media was changed after 6 hr. Cells were fixed 72 hr after the siRNA transfection and stained as described above. Images from three independent experiments were acquired using identical acquisition settings. To minimise effects of overexpression, only dim HA-CHMP7$^R$-transfected cells (area-normalised HA-fluorescence intensities below 250 AU; background of approximately 100 AU) were selected for quantification. For analysis of pSer3 CHMP7 staining, cells growing on glass coverslips were washed into transport buffer (20 mM HEPES pH 7.3, 110 mM potassium acetate, 5 mM sodium acetate, 2 mM magnesium acetate, 1 mM EGTA, 2 mM DTT and supplemented with protease and phosphatase inhibitors *Adam et al., 1990*) and containing 50 µg/mL digitonin (CAL-51) or 100 µg/mL digitonin (HeLa) as necessary. Cells were incubated on ice for 5 min, then were fixed in 1% PFA in transport buffer and moved to room temperature. Cells were permeabilised in transport buffer containing 0.5% Saponin and immunofluorescence proceeded as above.

## Live cell imaging

Cells stably expressing the indicated proteins, or edited to express fluorescent proteins, were plated in 4- or 8-chamberslides (Ibidi). Cells were transfected with the indicated siRNA where necessary. Cells were transferred to an inverted Nikon Eclipse Ti2 microscope teamed with an Andor Dragonfly 200 Spinning Disc Confocal unit and paired with Andor Zyla sCMOS and Andor iXon EM-CCD cameras, with attached environmental chamber (Oxolabs) and imaged live using 20x dry or 60x oil-immersion objectives, typically acquiring frames every 30 s. In all cases, 405, 488, or 561 laser lines were used for illumination. To enable multifield overnight imaging of GFP-CHMP7$^{\delta Helix6}$, Videos

were acquired using a x20 dry objective, preventing resolution of localisation to the ER or assembly in fine puncta (*Figure 3A*). For monopolar spindle assays, cells were incubated for 16 hr with S-trityl L-cystine (5 µM) or Nocodazole (50 ng/mL) to arrest cells in M-phase. Cells were released from M-phase through addition of the CDK1 inhibitor RO-3306 to a final concentration of 9 µM. Cells were either imaged live or were collected for biochemical experiments as described below.

## Immunoprecipitation

For extraction of GFP-CHMP7 for PhosTag SDS-PAGE, cells were lysed on ice in RIPA buffer (150 mM NaCl, 50 mM Tris-Cl pH 7.4, 1% NP-40, 0.2% SDS, 0.5% Sodium deoxycholate) supplemented with protease (HALT, Sigma) and phosphatase (PhosSTOP, Roche) inhibitor cocktails. Clarified lysates were incubated with GFP-trap magnetic agarose beads (Chromotek) for 1 hr, washed thrice using Pulldown Wash buffer (150 mM NaCl, 50 mM Tris-Cl pH 7.4, 0.1% NP-40) after magnetic capture, and released from the resin through boiling in 2 x Laemmli buffer. For biochemical fractionation, synchronised cells at the indicated timepoints were collected on ice, transferred into fractionation buffer (20 mM HEPES pH 7.4, 10 mM KCl, 2 mM $MgCl_2$, 1 mM EDTA, 1 mM EGTA, 1 mM DTT) supplemented with protease and phosphatase inhibitors and left to swell on ice for 15 min. Cells were lysed by 15 passages through a 27-gauge needle and left on ice for 20 min. Nuclei and extra-nuclear fractions were collected by centrifugation (4000 x g, 10 min), lysed in EDTA-free Laemmli buffer, sonicated and examined by PhosTag SDS-PAGE.

## GST-pull down assay

293T cells expressing GST-tagged constructs were lysed on ice in Pulldown Lysis buffer (150 mM NaCl, 50 mM Tris-Cl pH 7.4, 1% NP-40) supplemented with protease inhibitor (HALT, Sigma) cocktail. Clarified lysates were incubated by glutathione sepharose 4β beads (GE healthcare) for 2 hr, washed thrice using Pulldown Wash buffer after centrifugal capture, and released from the resin through boiling in 2 x Laemmli buffer.

## Statistical analysis

Two-tailed Student's T-tests, or ordinary one-way ANOVA with the indicated post-hoc tests were used to assess significance between test samples and controls and were performed using GraphPad Prism. N-numbers given as the number of independent experiments, n-numbers given as the number of cells analysed.

## Acknowledgements

JCG is a Wellcome Trust Senior Research Fellow, CLS received a BBSRC LIDO PhD studentship. This work was supported by the Francis Crick Institute which receives its core funding from Cancer Research UK (FC001002, FC001143), the UK Medical Research Council (FC001002, FC001143), and the Wellcome Trust (FC001002, FC001143). We thank the Crick Structural Biology and Advanced Light Microscopy Science Technology Platforms for access to instruments. This research was funded in whole, or in part, by the Wellcome Trust (206346/Z/17/Z, FC001002, FC001143). For the purpose of Open Access, the author has applied a CC BY public copyright licence to any Author Accepted Manuscript version arising from this submission.

## Additional information

### Funding

| Funder | Grant reference number | Author |
|---|---|---|
| Wellcome Trust | 206346/Z/17/Z | Jeremy G Carlton |
| Biotechnology and Biological Sciences Research Council | BB/M009513/1 | Caroline L Stoten |
| Francis Crick Institute | FC001002 FC001143 | Peter B Rosenthal Jeremy G Carlton |

The funders had no role in study design, data collection and interpretation, or the decision to submit the work for publication.

## Author contributions
Alberto T Gatta, Yolanda Olmos, Conceptualization, Investigation, Methodology, Writing - original draft, Writing - review and editing; Caroline L Stoten, Qu Chen, Investigation, Methodology; Peter B Rosenthal, Conceptualization, Supervision; Jeremy G Carlton, Conceptualization, Supervision, Funding acquisition, Investigation, Methodology, Writing - original draft, Project administration, Writing - review and editing

## Author ORCIDs
Alberto T Gatta (iD) https://orcid.org/0000-0002-2404-7351
Yolanda Olmos (iD) http://orcid.org/0000-0002-5952-1607
Peter B Rosenthal (iD) http://orcid.org/0000-0002-0387-2862
Jeremy G Carlton (iD) https://orcid.org/0000-0002-7255-1894

## Decision letter and Author response
Decision letter https://doi.org/10.7554/eLife.59999.sa1
Author response https://doi.org/10.7554/eLife.59999.sa2

# Additional files

## Supplementary files
- Source data 1. Lane crops of blots used in this manuscript.
- Source data 2. Scans of full blots used in this manuscript.
- Transparent reporting form

## Data availability
Source data files have been provided for Figure 1, Figure 1 Supplement 2, Figure 1 Supplement 3, Figure 1 Supplement 5, Figure 2, Figure 2 Supplement 1, Figure 2 Supplement 2, Figure 3, Figure 4, Figure 4 Supplement 1, Figure 5, Figure 5 Supplement 1, Figure 5 Supplement 2, Figure 5 Supplement 3, Figure 6 and Figure 6 Supplement 1.

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
