## [Decision Letter]

**Acceptance summary:**

The authors show that CDK1 phosphorylates Chmp7, after which Chmp7 is less prone to polymerize, most likely because its interaction with Lem2 is reduced. These results provide the first insight into how the cell cycle machinery can control the assembly of Chmp7 (and nuclear ESCRT-III) and how polymerization Chmp7-ESCRT-III assembly can be spatially restricted to the reforming nuclear envelope.

**Decision letter after peer review:**

[Editors' note: this paper was reviewed by Review Commons.]

Thank you for submitting your article "CDK1 controls CHMP7-dependent nuclear envelope reformation" for consideration by *eLife*. Your article has been reviewed by 3 peer reviewers, and the evaluation has been overseen by a Suzanne Pfeffer as the Senior and Reviewing Editor.

The reviewers have discussed the reviews with one another and the Reviewing Editor has drafted this decision to help you prepare a revised submission.

The manuscript 'Cdk1 controls Chmp7-dependent nuclear envelop reformation' by Jeremy Carlton and co-workers provides important new insight that helps to explain how the formation of ESCRT assemblies at the nuclear envelope (NE) is regulated. The authors have used a combination of live cell imaging, mutational analysis, and biochemical approaches to demonstrate how the interplay of Lem2 (an inner nuclear envelope membrane protein) and Chmp7 is coordinated during cell cycle progression. Because Chmp7 is an ESCRT-II/-III hybrid protein that nucleates ESCRT-III assemblies at the NE, its regulation by Cdk1 spatially and temporally restricts ESCRT -III assemblies to the reforming NE.

Specific revision requests for *eLife*:

1. Please address directly the in vivo role of Chmp7 phosphorylation for nuclear envelope integrity and morphology during interphase and after mitosis. It could be done by generating cell-lines in which endogenous Chmp7 was depleted and replaced with WT-Chmp7 (rescue) or phospho-mimetic- or non-phosphorylated Chmp7 mutants. In these cell-lines, they should analyze Lem2 clustering as well as nuclear integrity using GFP-NLS.

2. The Chmp7 phosphorylation paradox needs to be addressed somehow in the revision: Phosphorylation of CHMP7 restricted its polymerisation and yet CHMP7 was observed to assemble at the reforming nuclear envelope during the time period that it was phosphorylated. Perhaps one way to address this question in vivo is to generate cells that express at the same time GFP-Chmp7 S3A, S441A and phosphomimetic RFP-Chmp7 S3D, S441D and then follow their distribution in cells over time using live cell imaging. One would expect to see a clear separation of the two pools during mitosis.

3. Why do LEM2 clusters persist in CHMP7 depleted cells? Please check VPS4 localization in CHMP7 depleted cells to measure whether it localises to LEM2 clusters.

4. Please test in-vitro binding between Lem2 and the Chmp7 phospho mutants.

5. It would be great if possible, but not essential, to visualise how phosphorylation impacts CHMP7's oligomerization status using AFM or, more easily, negative stain electron microscopy, comparing with published control conditions (full-length versus ESCRT-III domain alone, in the presence of model membranes, versus in the presence of the C-terminal WH domain of LEM2). Can the observations be reconciled with the paradoxical monopolar spindle data? Specifically, these monopolar spile data led the authors to claim that CHMP7 is phosphorylated during the critical window when CHMP7 assembly activity is required for envelope sealing. As the authors note, this is hard to reconcile with their data showing that phosphorylation enhances solubility and that phospho-dead mutants assemble avidly with LEM2 and downstream ESCRT-IIIs, like IST1. At minimum please modify text accordingly.

---

## [Author Response]

Specific revision requests for eLife:1. Please address directly the in vivo role of Chmp7 phosphorylation for nuclear envelope integrity and morphology during interphase and after mitosis. It could be done by generating cell-lines in which endogenous Chmp7 was depleted and replaced with WT-Chmp7 (rescue) or phospho-mimetic- or non-phosphorylated Chmp7 mutants. In these cell-lines, they should analyze Lem2 clustering as well as nuclear integrity using GFP-NLS.

We have performed new experiments to address this function by generating new cell lines expressing GFP-NLS, LEM2-mCh. Despite repeated attempts, we were unable to generate triply-stable lines bearing phosphomimetic versions of CHMP7, LEM2 and a reporter of nuclear integrity. As such, we turned to a carefully controlled transient rescue of the CHMP7-depletion phenotype employing RNAi-resistant HA-CHMP7 constructs and in these lines analysed both LEM2 clustering and nuclear integrity. These data revealed that the LEM2-clusters formed in CHMP7 depleted cells compromised NE integrity. These clusters, and the NE-integrity defect, could be rescued by HA-CHMP7^R^, but not by HA-CHMP7^R^ S3A/S441A. The new data is presented in Figure 1H and 1I, Figure 6A, 6B and Figure 6 Supplement 1A.

When performing these experiments, we were surprised to find that both HA-CHMP7^R^ S3D/S441D and HA-CHMP7^R^ S3E/S441E were able to rescue the clustering of LEM2 and the NE-integrity defect. We were intrigued by these findings and went on to examine the behaviour of GFP-CHMP7 S3D/S441D during M-exit. We discovered that it was still able to display limited assembly at the reforming NE, albeit with recruitment being significantly delayed and being significantly less intense (new Figure 5F-5H). We also performed new in-vitro experiments (see point 4 below) that show that phosphorylation or phosphomimetic residues at these positions could suppress CHMP7 sedimentation, its interaction with LEM2 and its polymerisation (new Figure 4E-4H, Figure 4 Supplement 1C and 1D. We note that this restriction on assembly in-vitro is partial, and we hypothesise that this allows it to be overridden by the high concentrations of LEM2 assembling at the reforming NE. We incorporate this into our discussion and also make comment that this may be evidence that alternate polymerisation stimuli may exist.

To address directly the physiological role of CHMP7-dependent LEM2 cluster dissolution, we analysed markers of DNA damage in these cells. We show in the revised manuscript that 53BP1 and gH2AX foci are associated with LEM2 clusters (new Figure 1 Supplement 5), demonstrating their detrimental effect on cellular physiology.

2. The Chmp7 phosphorylation paradox needs to be addressed somehow in the revision: Phosphorylation of CHMP7 restricted its polymerisation and yet CHMP7 was observed to assemble at the reforming nuclear envelope during the time period that it was phosphorylated. Perhaps one way to address this question in vivo is to generate cells that express at the same time GFP-Chmp7 S3A, S441A and phosphomimetic RFP-Chmp7 S3D, S441D and then follow their distribution in cells over time using live cell imaging. One would expect to see a clear separation of the two pools during mitosis.

This is an important point, and we apologise for not making our argument clear in the initial submission. The reviewers are correct in that phosphorylation of all the CHMP7 in the cell persists beyond the time period in which assembly at the NE occurs. In the original submission, we hypothesised that the NE-assembling pool was dephosphorylated in advance of the pool in the peripheral ER/cytosol, but we did not test this. Here, we have performed new experiments employing biochemical fractionation of the chromatin-associated and non-chromatin associated pools at defined times during a synchronised M-exit, and combined this with analysis of the phosphorylation status of CHMP7 using PhosTag SDS-PAGE. This revealed firstly that at all time points, only a minor fraction of CHMP7 was recoverable in the chromatin associated pool (new Figure 5 Supplement 2A-2C), suggesting that only a minor fraction of CHMP7 is involved in polymerisation at the reforming NE. This is consistent with our imaging approaches which demonstrate CHMP7 polymerisation is limited to the minor pool in direct contact with the NE. By analysing the rate of dephosphorylation of these two pools, we find that the chromatin-associated pool is dephosphorylated in advance of the pool in the peripheral ER/cytosol (new Figure 5 Supplement 2A-2C). We suggest that persistent phosphorylation of the ER-pool acts to restrict inappropriate assembly and capture of residual LEM2 on the peripheral ER. Consistent with this, we observe inappropriate clusters of GFP-CHMP7^R^ S3A/S441A and LEM2-mCh forming on the ER during M-exit (new Figure 6C). We suggest that advanced dephosphorylation of a chromatin-associated pool of CHMP7 licenses assembly at the reforming NE.

To strengthen these observations, we generated a new phospho-specific antibody against CHMP7 Ser3. This antibody is characterised in Figure 5 Supplement 2D-2I. We note that the ER is notoriously hard to examine by fixation and immunofluorescence, but after careful optimisation, we were able to demonstrate this antibody’s ability to detect an siRNA-sensitive signal of phosphorylated CHMP7 on the mitotic ER. Under identical staining conditions, this antibody did not illuminate the pool of CHMP7 assembling at the reforming NE. These data are presented in new Figure 5 Supplement 2D-2I,, and we hope support our hypothesis that the pool of CHMP7 assembling at the reforming NE is dephosphorylated in advance of the major pool in the peripheral ER.

3. Why do LEM2 clusters persist in CHMP7 depleted cells? Please check VPS4 localization in CHMP7 depleted cells to measure whether it localises to LEM2 clusters.

We thank the reviewers for raising this – this is an interesting point that provides insight into how the NE clusters of LEM2 might get disassembled. We have generated new cell lines stably expressing LEM2-mCh and VPS4A-L-GFP. We find that VPS4 neither localises to the reforming NE in the absence of CHMP7 (new Figure 1 Supplement 4B), nor does it localise to the clusters of LEM2 formed in the absence of CHMP7 (new Figure 1 Supplement 4C). As expected from the literature, these data indicate that VPS4 is recruited to the reforming NE downstream of CHMP7. Interestingly, in the absence of IST1, VPS4 recruitment to the reforming NE was normal, and as LEM2 clusters at the NE did not form under these conditions, these data suggest recruitment of VPS4 correlates with dissolution of LEM2 clusters, indicating that this may be a consequence of canonical ESCRT-III activity and, importantly, to be independent of IST1.

4. Please test in-vitro binding between Lem2 and the Chmp7 phospho mutants

We thank the reviewers for suggesting these experiments and believe they have added useful mechanistic insight into the behaviour of CHMP7. We began by extending our sedimentation assay that demonstrated co-sedimention of HA-LEM2^CT^ with CHMP7 as a proxy for the CHMP7-LEM2 interaction. We generated recombinant versions of full length CHMP7 bearing phosphomimetic residues at Ser3 and Ser441 and showed that in-vitro, these phosphomimetic residues reduced sedimentation of CHMP7, suggesting that these changes impaired the self-association necessary for clustering and sedimentationn. The reviewers are correct in pointing out that we cannot know the polymerisation state of the sedimented material, and we have been careful to reword this as evidence of CHMP7 ‘clustering’. Under these conditions, co-sedimentation of HA-LEM2^CT^ with CHMP7 S3D/S441D or CHMP7 S3E/S441E was reduced, suggesting that phosphorylation of CHMP7 impairs its ability to interact with LEM2. These data are presented in new Figure 4C and 4D.

However, as Reviewer 2 pointed out in their Review Commons review, ‘CHMP7 alone may form sticky aggregates, which will also pellet in sedimentation’. We wanted to repeat these experiments using a centrifugation-independent interaction assay to be sure that we were examining the interaction, rather than just the ability of CHMP7 to sediment and capture LEM2 non-specifically. As such, we covalently coupled anti-CHMP7 antisera to magnetic beads and performed the CHMP7/LEM2 interaction assay in-vitro. Here, we found that CHMP7 S3D/S441D’s interaction with LEM2^CT^ was significantly reduced (new Figure 4E-4H). We extended these data to show that the interaction of WT CHMP7 with LEM2^CT^ could be suppressed by directly phosphorylating CHMP7 using CDK1 (new Figure 4E – 4F). We thank the reviewer for these insightful suggestions and conclude that phosphorylation of CHMP7 on Ser3 and Ser441 suppresses its ability to bind LEM2.

5. It would be great if possible, but not essential, to visualise how phosphorylation impacts CHMP7's oligomerization status using AFM or, more easily, negative stain electron microscopy, comparing with published control conditions (full-length versus ESCRT-III domain alone, in the presence of model membranes, versus in the presence of the C-terminal WH domain of LEM2). Can the observations be reconciled with the paradoxical monopolar spindle data? Specifically, these monopolar spile data led the authors to claim that CHMP7 is phosphorylated during the critical window when CHMP7 assembly activity is required for envelope sealing. As the authors note, this is hard to reconcile with their data showing that phosphorylation enhances solubility and that phospho-dead mutants assemble avidly with LEM2 and downstream ESCRT-IIIs, like IST1. At minimum please modify text accordingly.

We really wanted to test this, and as these were new techniques for my lab, access to equipment, training and assay optimisation took some time. Using negative-stain electron microscopy, we can now report that in the presence of LEM2^CT^, we observe the generation of a polymer of CHMP7 that matches well the published dimensions and periodicity from von Appen et al., 2020. Performing these assays using CHMP7 S3D/S441D, we were unable to identify any polymeric assemblies, and could observe only monomers and aggregates. Given our interaction data suggesting that the phosphorylated version of CHMP7 is impaired in its ability to bind LEM2, we think it likely that this underlies its failure to polymerise in-vitro. We have included this data in the manuscript (Figure 4I and Figure 4 Supplement 2). In response to the paradoxical monopolar spindle data, we hope that our explanation and biochemical data have assuaged the reviewer’s concerns, as outlined in point 2. For assistance, performance and interpretation of negative stain electron microscopy, we have included Qu Chen and Peter Rosenthal as co-authors.

[Editors' note: we include below the reviews that the authors received from another journal, along with the authors’ responses.]

Reviewer #1 (Evidence, reproducibility and clarity (Required)):The manuscript 'Cdk1 controls Chmp7-dependent nuclear envelop reformation' by Jeremy Carlton and co-workers provides important new insight that helps to explain how the formation of ESCRT assemblies at the nuclear envelop (NE) is regulated. The authors have used a combination of live cell imaging, mutational analysis, and biochemical approaches to demonstrate how the interplay of Lem2 (an inner nuclear envelope membrane protein) and Chmp7 is coordinated during cell cycle progression. Because Chmp7 is an ESCRT-II/-III hybrid protein that nucleates ESCRT-III assemblies at the NE, its regulation by Cdk1 spatially and temporally restricts ESCRT -III assemblies to the reforming NE.

We thank the reviewer for their positive assessment of our work.

Major comments:The major conclusion are convincing. In particular the authors show that Cdk1 phosphorylates Chmp7 at mitotic entry on two sites (S4A, S441). Cdk1 dependent phosphorylation prevents Chmp7 activation/polymerization and phosphorylated Chmp7 can no longer engage Lem2 at the peripheral ER and thus limits ESCRT -III assemblies at the NE to mitotic exit. Defects in this process trigger unscheduled Lem2-Chmp7-ESCRT-III assemblies at the peripheral ER during mitotic exit and lead to the formation of Lem2 aggregates. The overall conclusions of the paper are important and provide a significant step forward in our understanding how the ESCRT machinery is controlled by the cell cycle machinery to re-establish NE integrity in interphase and after mitosis. Some points require clarification – I believe that most comments can be addressed without further experiments:– What is the functional consequence of failure to phosphorylate Chmp7 for NE integrity and morphology during interphase and after mitosis?– How do Lem2 cluster comprise NE function, morphology an integrity?

This is an interesting and important question and the reviewer is right that we didn’t fully explore this in our original submission. I hope that these points have been addressed in the response to *eLife* review, Point 1.

– Data for Chmp7 and Lem2 knock down (KD) efficiency should be provided, ideally also rescue experiments should be performed.

This is important, we have now provided western blotting data to validate the knockdown efficiency achieved with all siRNAs used in the manuscript (new Figure 1B). We have also performed siRNA rescue experiments involving both HA-CHMP7^R^ constructs and stably expressed GFP-CHMP7^R^ phosphomutants to address the role of phosphorylation. I hope these points have been addressed in the response to *eLife* review, Point 1.

– The heterochromatic HP1 staining looks much brighter with a different pattern (sometimes more peripheral, perhaps co-localizing with Lem2) in the Chmp7 depleted cells. Does that imply that Chmp7 negatively regulates the formation of heterochromatin in general?

This is an interesting point, and something we had not picked up on before. We have revisited our imaging and have quantified the fluorescence intensities of the non-clustered HP1 signal in Control and RNAi-treated cells, which revealed neither consistent nor significant elevation of this signal in depleted cells (Author response image 1). Whilst the link between clustered and non-clustered HP1 is certainly interesting, we don’t believe it is central to the message of this manuscript. Additionally, whilst there are a few peripheral HP1 clusters remaining in CHMP7 depleted cells, careful analysis reveals that they juxtapose, but do not colocalise with, LEM2 clusters. We provide an additional panel (Figure 1 Supplement 5E) that demonstrates this juxtaposition, and discuss this in the revised main text on page 3.

**Author response image 1. sa2fig1:** HP1 fluoresence intensity in siRNA transfected cells N = 3, n = 262 (control), 233 (CHMP7 si-1), 254 (CHMP7 si-2), 193 (IST1 siRNA).

We also wondered if these LEM2 clusters were damaging for cells. To do this, we analysed markers of DNA damage in these cells. We show in the revised manuscript that 53BP1 and gH2AX foci are associated with LEM2 clusters, demonstrating their detrimental effect on cellular physiology. These data are presented in Figure 1 Supplement 5A.

– The results show that Cdk1 phosphorylation prevents unscheduled Chmp7 polymerization (Figure 3 and Figure 4). Yet at the same time the authors argue that 'ESCRT-III dependent NE formation occurs at a time period when Chmp7 is globally phosphorylated and that Chmp7 dephosphorylation occurs after nuclear envelop reformation'. Please clarify how that goes together? What am I missing?

We apologise to the reviewer for our clumsy wording. A caveat of the biochemical assays in the original submission is that they reported the phosphorylation status of all of the CHMP7 in the cell, whereas the imaging demonstrates that only a limited pool of CHMP7 in proximity to the nuclear envelope polymerises. We hope that our biochemical fractionation studies and phospho-specific antibody generation have resolved this paradox by demonstrating that the NE-pool of CHMP7 is dephosphorylated in advance of the cytosolic and ER pool. Please see also response to *eLife* review, Point 2.

Reviewer #1 (Significance (Required)):The role of the ESCRT machinery in NE biology is an important and hot topic, with many papers published recently in peer reviewed journals and on bioRvix. The overall conclusions of this paper are important and provide a significant step forward in our understanding how the ESCRT machinery is controlled by the cell cycle machinery to re-establish NE integrity in interphase and after mitosis. Hence this paper will be of interest to a broad readership.My field of expertise is the ESCRT machinery.Reviewer #2 (Evidence, reproducibility and clarity (Required)): The new study from the Carlton lab focuses on the determinants of nuclear membrane homeostasis in human cell mitosis. Their particular focus is on the conserved and now established interaction between the inner nuclear membrane protein, LEM2, and the ESCRT-II/III protein, CHMP7. These proteins are segregated from each other by the nuclear envelope during interphase, and function together during late mitosis when the nuclear membrane needs to reform or during interphase if the nucleus is damaged. In their new work, these authors provide high-quality imaging data and some biochemistry aimed at understanding how LEM2 and CHMP7 activities are regulated. Their study convincingly establishes that CHMP7s activity pattern depends on phosphoregulation, adding another level of ESCRT regulation during mitosis. Specifically, their data show that CHMP7 is phosphorylated in early mitosis by CDK1, following mitotic patterns. CHMP7 is also phosphorylated by CDK1 in vitro, and this appears to increase CHMP7's solubility. Phospho-dead mutant constructs of CHMP7 (Ser to Ala) accumulate in mislocalized puncta or "clusters" upon the ER, where they also inappropriately trap another downstream ESCRT-III protein, IST1, preventing it from reaching the nascent inner nuclear membrane. These clusters are mislocalized because previous research has established that LEM2 and CHMP7 usually cluster within gaps in the nascent nuclear envelope. The aberrant mutant clusters trap other ESCRT-IIIs proteins, like IST1, as well as LEM2-and this prevents LEM2 from reaching the inner nuclear envelope.In general, the work is of high quality, especially the imaging experiments. Specific biochemical results, however, are more ambiguous. In many cases, careful rephrasing will suffice to help readers recognize caveats in interpretation but certain claims, which I will expand upon point-by-point, should be softened or will require additional experiments.

We thank reviewer 2 for their positive assessment of our work.

Major Comments: 1. The results from the monopolar spindle assays confused me, and given the highly artificial setting induced by these drugs are of unclear value to the current report. These data led the authors to the paradoxical claim that CHMP7 is phosphorylated during the critical window when CHMP7 assembly activity is required for envelope sealing. But as the authors note, this is very hard to reconcile with their data showing that phosphorylation enhances solubility and that phospho-dead mutants assemble avidly with LEM2 and downstream ESCRT-IIIs, like IST1. I recommend removing these data, even though the authors also use data from the monopolar spindle assays to support their claim that a "microtubule network is dispensable for ESCRT-III assembly at the reforming nuclear envelope." This claim already enjoys support from prior studies documenting ESCRT-III's role in repairing the nuclear envelope during interphase when mitotic spindle fibers are absent.

We apologise to the reviewer for our clumsy wording. A caveat of the biochemical assays in the original submission is that they reported the phosphorylation status of all of the CHMP7 in the cell, whereas the imaging demonstrates that only a limited pool of CHMP7 in proximity to the nuclear envelope polymerises. We hope that our fractionation studies and phospho-specific antibody generation have resolved this paradox by demonstrating that the NE-pool of CHMP7 is dephosphorylated in advance of the cytosolic and ER pool. Please see also response to *eLife* review, Point 2.

2. The term "polymerization" needs to be qualified in the text, or data revealing the existence of polymers need to be included. The clusters in the ER observed by light microscopy may contain polymerized CHMP7, but I suggest a more open phrasing (e.g. clustering) or, if the authors intend to dissect this further, additional experimental evidence. For example, does CHMP7-dHelix6 polymerizes in vitro? In addition, the authors use SDS-Page coupled to centrifugation to report CHMP7 pelleting as a proxy for polymerization (Figure 2F). Whether full-length CHMP7 alone polymerizes under these in vitro conditions has not been established, the cited research (ref #6) relied on liposomal membranes or the LEM2 winged helix domain to trigger CHMP7 polymerization. This is important because CHMP7 alone may form sticky aggregates, which will also pellet in sedimentation and SDS-Page based assays-especially after a 60 min incubation at 30{degree sign}C. This experiment and the associated figure panel are not critical to the central message of CDK1-mediated phosphoregulation. If the authors want to dissect CHMP7 polymerization further, I suggest to include visualization by AFM or EM and controls based on published conditions (CHMP7 + liposomes or CHMP7 + LEM2-CTD). Such experiments would also enable the authors to visualize whether and how whether CDK1-phosphorylated CHMP7 polymerizes.

We thank the reviewer again for their thoughtful suggestions. We hope that the new data described in response to *eLife* review, Point 5 has addressed this issue.

3. The persistent LEM2 clusters in the absence of CHMP7 are intriguing, especially given the observations in fission yeasts like pombe and japonicus that suggest Vps4-mediated disassembly is a critical aspect of nuclear homeostasis. Does it seem the simplest explanation is that the loss of CHMP7 precludes the recruitment of downstream ESCRT-III proteins, like CHMP4 and other CHMPs? Without this sequential cascade, VPS4-family ATPases are never recruited?

This is a good point. We visualised VPS4 in CHMP7-depleted cells exhibiting LEM2 clusters and, as the reviewer suggested, demonstrated that it was no longer recruited. Please see also the response to *eLife* review, Point 3.

Minor Comments:4. The idea that phosphorylation regulates the dynamics of mitosis is well established, and when looking at public databases, it's clear that both CHMP7 and LEM2 have many annotated sites of phosphorylation (Serine/Tyrosine/Threonine). A broader discussion (either in the introduction or the discussion) about phosphoregulation would add value. Phosphorylation also regulates IST1 activity and LEM-BAF interactions, for example.

We thank the reviewer for this important point, this will help accessibility for more generalist readers. We have included more discussion of how phosphorylation can regulate the biology of organelle remodelling complexes during mitosis in the introduction and Discussion sections, with relevance for ESCRT systems highlighted.

5. In the discussion the authors write "Importantly, LEM2 assembly at the reforming nuclear envelope occurred in the absence of CHMP7, suggesting that LEM2 is upstream and does not form as part of the ESCRT-III polymer. However, consistent with recent reports from S. japonicus [16,17], LEM2's failure to be disassembled in the absence of CHMP7 suggests that CHMP7 acts to decluster and proper release of LEM2, allowing it to correctly populate the INM. Our biochemical data indicate that polymeric CHMP7 can capture LEM2, raising the possibility that this INM protein may instead be a 'cargo' that is actively sorted onto the INM through the activity of CHMP7 at sites of annular fusion." This is a creative but speculative notion that stretches an analogy to other sites of ESCRT activity, and stretches it too far for my taste. What does it mean that LEM2 does not form as part of the ESCRT-III polymer? The constituents and properties of the ESCRT-III polymer in question remain incompletely understood-and the LEM2 winged-helix domain has been reported to co-assemble with CHMP7 around spindle fibers. Why and how should CHMP7 "sort" LEM2 to sites of annular fusion? Based on what data? Several other studies, including data in this manuscript (Figure 1B), show that LEM2 is required to recruit CHMP7 to the nuclear envelope, that LEM2 localization depends on the LEM-BAF interaction and BAFs ability to coat the chromatin disc. Further, when LEM2 is depleted CHMP7 does not cluster.

The reviewer is right in that we took a little artistic license in the Discussion section, and we have reined in our speculation here. We have also amended the text to remove reference to LEM2 incorporation into the ESCRT-III polymer, but think it is important to note that LEM2 polymerisation dynamics were unperturbed by CHMP7 depletion.

6. Please label the micrographs in Figure S1A.

Done.

7. Figure S3J is missing (phosphatase treatment).

We apologise for this and considered re inserting the relevant figure. However, these inhibitors are blunt tools and the biology relating to CHMP7 dephosphorylation is likely complex. We didn’t consider that these data made significant contribution to the manuscript and will look instead to explore CHMP7 dephosphorylation in future work.

8. There are a few missing periods, ratio misspelled as ration, etc.

We apologise for this and have rechecked the document for typos.

Reviewer #2 (Significance (Required)): Nuclear reformation and repair are essential processes because defects lead to phenotypes ranging from progeria to tumorigenesis. The current study builds on foundational work, including a key 2015 paper from the Carlton lab that demonstrated a role for the ESCRTs in sealing the anaphase nuclear envelope.I am an expert in LEM2 and ESCRT-III protein structure and function. Among other contributions, our lab discovered that LEM2 serves as a nuclear site-specific adaptor for the ESCRTs by recruiting CHMP7 directly.Reviewer #3 (Evidence, reproducibility and clarity (Required)): The bulk of the conclusions derive from outstanding microscopy of siRNA depleted HeLa cells in which the authors have expressed variants of CHMP7 and LEM2. This manuscript is an excellent contribution to the field of understanding the dynamics of the nuclear membrane during mitosis. Critically, the concept that microtubules have a key role in this is effectively challenged by the work. The key conclusions are convincing and fairly straightforward to follow. The experimental methods seem to be complete, and it should be possible to replicate the results. The manuscript is succinct and makes a straightforward conclusion as to the role of CDK1. There are no major flaws or missing controls, within the scope of the manuscript. The statistical analyses appear to be complete and appropriate.There are a few minor points that could make the manuscript easier to follow. In many places there is jargon used that is probably appropriate for the field, but obscures understanding by a more general readership.

We thank the referee for their positive assessment of our work. We have made attempts to simplify the text and hope that the jargon is reduced, or better explained

1. The authors immediately refer to LEM2 as an INM protein in the abstract. It should be introduced as an inner nuclear membrane (INM) protein.

We apologise for this and have ensured to define this and all acronyms before using them.

2. There are multiple references to "annular fusion". Given that there is nothing in the manuscript that remotely resembles an annulus, it might not be all that helpful for the authors to use this jargon.

Annular fusion is a term coined by Burke and Ellenberg in 2002 to describe the topologically unique process of nuclear envelope sealing that was later found to be performed by the ESCRT machinery. However, we realise that this term is not particularly accessible for a generalist reader and we have removed this jargon from the manuscript.

3. The authors refer to elements of the CHMP7 and LEM2 anatomy (LEM domain, chromatin tethering region, CHMP-binding region, helix5, helix6 etc), so it would be helpful of they could include a domain organisation scheme (a bar diagram) in the corner of a figure or in the supplementary material.

This is a nice idea, we have included a schematic (Figure 1A) to help orient readers.

4. The notations in Figure S2 are confusing. S2B has 377-CT, α6:LLL-AAA and 377-CT, α5:LLL-AAA +α6:LLL-AAA, yet other figures and text refer to these as GFP-CHMP7NES- and GFP-CHMP72xNES-. The authors should change the names in S2B or list both labels in S2B.

We apologise to the reviewer for this jumbled mark-up – we should have done a better job at being consistent throughout the manuscript and figure labelling. We have now relabelled these consistently and explained in the legend what the abbreviated notation refers to.

5. It is not clear how Figure S4D relates to S4C, though the legend says that it is related. The text states that long-term culture of cells bearing GFP-CHMP7R S3A S441A resulted in the loss of LEM2 from the INM with its retention in singular cytoplasmic clusters. What is confusing is that the caption appearing on the second clip in video 15 is apparently incorrect. The caption says it is CHMP7R-GFP, when it should be GFP-CHMP7R S3A S441A.

Thanks for spotting this, we have corrected it for the resubmission. The text and figures have been reorganised and we hope properly called out.

6. In the Figure S3E legend, there is the statement "Data were quantified in Figure 3G", however, 3G has no quantitation. This should be Figure 3F.

We apologise for the mislabelling and have corrected it for the resubmission. Thank you for spotting this!

7. In the Discussion section on p. 6 "decluster and proper release of LEM2," should be "decluster and properly release".

Thanks for spotting this – we will correct.

Reviewer #3 (Significance (Required)):The authors have shed new light on the mechanism of nuclear envelop resealing after mitotic exit. Previous observations by the authors and others showed that the ESCRT-III-like protein CHMP7 has an important role in this process. It was previously reported that the inner nuclear membrane protein LEM2 is involved in the process of polymerising CHMP7, and that this transient polymerisation is important for the reformation of the nuclear envelope. The current manuscript addresses a crucial question: How can CHMP7 be excluded from the nucleus by a NES during the period of the cell cycle in which the nuclear membrane has disassembled and interspersed with ER membranes? The authors convincingly show that during the period when the nuclear membrane is not present, sorting of the cytoplasmic CHMP7 away from the INM LEM2 is accomplished through a classic cell-cycle regulator, CDK1. When CHMP7 is phosphorylated by CDK1 at S3 and S441, it prevents CHMP7 from inappropriately polymerising and mis-localising LEM2 to ER compartments.